# Defunctionalizing intracellular organelles such as mitochondria and peroxisomes with engineered phospholipase A/acyltransferases

Satoshi Watanabe [1,2,6] ✉, Yuta Nihongaki [1,2], Kie Itoh[1,2,3], Toru Uyama[4], Satoshi Toda [5], Shigeki Watanabe [1,2,3] & Takanari Inoue [1,2] ✉

Organelles vitally achieve multifaceted functions to maintain cellular homeostasis. Genetic and pharmacological approaches to manipulate individual organelles are powerful in probing their physiological roles. However, many of them are either slow in action, limited to certain organelles, or rely on toxic agents. Here, we design a generalizable molecular tool utilizing phospholipase A/acyltransferases (PLAATs) for rapid defunctionalization of organelles via remodeling of the membrane phospholipids. In particular, we identify catalytically active PLAAT truncates with minimal unfavorable characteristics. Chemically-induced translocation of the optimized PLAAT to the mitochondria surface results in their rapid deformation in a phospholipase activity dependent manner, followed by loss of luminal proteins as well as dissipated membrane potential, thus invalidating the functionality. To demonstrate wide applicability, we then adapt the molecular tool in peroxisomes, and observe leakage of matrix-resident functional proteins. The technique is compatible with optogenetic control, viral delivery and operation in primary neuronal cultures. Due to such versatility, the PLAAT strategy should prove useful in studying organelle biology of diverse contexts.

Membrane phospholipids help cellular organelles acquire distinct properties in their morphology and functionality[1,2]. These membranes not only serve as a physical partition from the cytosol and other organelles, but also become a platform for assembly of functional molecules to play pivotal roles in sustaining diverse activities of eukaryotes. There have been experimental approaches to perturb individual organelles. Genetic modification and pharmacological intervention target components in the biogenesis or functional pathways. Despite the utility, their actions are often limited by being either slow in time to allow for compensatory responses in cells, or specific for certain organelles and cannot be readily generalized to other organelles. Other approaches include the use of photosensitizer proteins such as KillerRed[3], SuperNova[4], and miniSOG[5,6] that generate reactive oxygen species (ROS), which oxidize surrounding biomolecular components upon light irradiation, leading to defunctionalization of target organelles. Despite their potential in universal application to different organelles, the mode and specificity of action of generated ROS need to be taken into consideration in interpreting

[1]Johns Hopkins University School of Medicine, Department of Cell Biology, Baltimore, MD 21205, USA. [2]Johns Hopkins University School of Medicine, Center for Cell Dynamics, Baltimore, MD 21205, USA. [3]Johns Hopkins University School of Medicine, Department of Neuroscience, Baltimore, MD 21205, USA. [4]Department of Biochemistry, Kagawa University School of Medicine, 1750-1 Ikenobe, Miki, Kagawa 761-0793, Japan. [5]WPI Nano Life Science Institute, Kanazawa University, Kanazawa 920-1192, Japan. [6]Present address: Laboratory of Protein Synthesis and Expression, Institute for Protein Research, Osaka University, 3-2 Yamadaoka, Suita, Osaka 565-0871, Japan. ✉e-mail: s-watanabe@protein.osaka-u.ac.jp; jctinoue@jhmi.edu

experimental results. Another technique utilizes channelrhodopsin which was targeted to organelle membranes to modulate membrane potential[7]. This strategy is exclusively suited for mitochondria that rely on the membrane potential for their functionality.

Due to the fundamental importance of the membranes for organelles to be functional, it is thought that techniques to alter membrane integrity should be effective to investigate the biological significance of individual organelles. With this in mind, we recently engineered actin nucleation promoting factors to generate constrictive force against target organelles for their deformation[8]. Interestingly, when the technique was applied to mitochondria, their major functions such as ATP synthesis were not significantly altered[8,9]. This suggested that induction of deformation is not necessarily sufficient for functional perturbation, and thus radical approaches such as striking remodeling of the membrane composition should be explored.

Phospholipase A (PLA) catalyzes hydrolysis of a phospholipid ester bond at *sn-1* (PLA$_1$) or *sn-2* (PLA$_2$) position and generates a lysophospholipid and a free fatty acid[10]. Growing evidence has illuminated a close relationship between PLA activity and morphological change of phospholipid membranes. Purified PLA proteins such as PLA$_1$, PLA2G4A, PLA$_2$ from bee venom and transmembrane PLA from bacteria, induced membrane bending, budding, tubulation and pearling in model membranes such as giant vesicles and supported phospholipid bilayers[11–14]. Furthermore, PLA$_1$ from snake venom not only swelled isolated mitochondria but also perturbed their functions[15], while inhibition of PLA$_2$ was reported to associate with tubulation of Golgi and trans Golgi network[16]. Of note, these PLAs need a supplementary condition such as non-physiologically high Ca$^{2+}$ concentration for the full catalytic activity. In contrast, one PLA$_1$/A$_2$ family member, phospholipase A/acyltransferase 3 (PLAAT3), has no requirement of such supplementary condition[17–19], and was reported to degrade organelle membranes during the development of lens transparency[20]. When a murine PLAAT3 truncation mutant that localizes to mitochondria was constitutively expressed in cultured cells, a series of mitochondrial defects were observed after two days, including swelling, fragmentation, membrane degradation, and loss of internal structures such as cristae[20].

To develop a molecular tool for organelle defunctionalization that is rapidly inducible, generalizable and target-specific, we saw a strong potential in the PLAAT family members[19]. We therefore decide to implement PLAAT proteins in chemically and optically inducible protein dimerization paradigms to reorganize phospholipids of individual organellar membranes. With rational optimizations, we demonstrate that these tools can successfully defunctionalize mitochondria and peroxisomes. The PLAAT molecular tool is genetically encoded, scalable, and also modularly designed, thus facilitating studies on physiological roles of organelles in various cellular processes.

## Results

### Chemically inducible translocation of PLAAT family proteins can deform mitochondria through phospholipase activity

In order to test the concept of utilizing human PLAAT family members for manipulation of morphology and function of organelles, we first chose one of the well-studied members, PLAAT3 (also known as AdPLA, HRASLS3, HREV107, PLA2G16). We then designed an experiment to achieve rapid recruitment of a full-length PLAAT3 to a target organelle using a chemically inducible dimerization (CID) system (Fig. 1a). Since a previous in vitro study showed that PLA deforms isolated mitochondria[15], we chose mitochondria as a model organelle. To induce translocation of PLAAT3 from the cytosol to the mitochondria surface in living cells, we used the FKBP-FRB dimerizing unit which rapidly and strongly interacts with each other in the presence of rapamycin[21]. We constructed two plasmids (Fig. 1b, top), PLAAT3 fused N-terminally to mCherry-FKBP (mCherry-FKBP-PLAAT3) and CFP-FRB fused C-terminally to the mitochondria outer membrane-targeting

sequence of monoamine oxidase A (CFP-FRB-MoA)[22]. These two constructs were co-transfected into COS-7 cells, which were then subjected to rapamycin added at 100 nM during fluorescence live-cell imaging. Within 30 min of rapamycin treatment, mitochondria were fragmented and swollen (Fig. 1b, upper panels, and Supplementary Movie 1). To examine whether this morphological change depends on the phospholipase activity, we repeated the experiment with a lipase-dead (LD) mutant of PLAAT3 where the Cys-113 residue that is critical for the catalytic activity is mutated to serine[23–25]. With mCherry-FKBP-PLAAT3-LD, there was no detectable morphological change in mitochondria (Fig. 1b, lower panels, Supplementary Movie 2). These results showed that inducible translocation of PLAAT3 to mitochondria membranes can rapidly alter overall morphology through phospholipase activity.

In humans, PLAAT family members consist of the following domains: Proline-rich (Pro) domain in the N-terminus, lecithin-retinol acyltransferase (LRAT) domain, putative transmembrane (TM) domain and C-terminus region in the most C-terminal end (CT) (Supplementary Fig. 1a). The generally conserved sequences among the PLAAT family motivated us to investigate whether members other than PLAAT3 possess the ability of inducing mitochondria deformation when used in the same translocation scheme described for PLAAT3. As a result, we found PLAAT1, 2 and 4, but not PLAAT5, induced mitochondrial deformation comparable to PLAAT3 (Supplementary Fig. 1b). A major difference between PLAAT5 and other members is a predicted TM domain; SOSUI and TMHMM do not predict PLAAT5 to contain a TM domain at the C-terminus (Supplementary Fig. 1a). To test if the TM domain is required for the organelle deformation, we mitochondrially targeted PLAAT4 with the truncated TM (PLAAT4-dTM) which still retains the phospholipase activity[26]. The truncated PLAAT4 did not induce organelle deformation (Supplementary Fig. 1b). These results suggested that, besides the catalytic activity, the TM is crucial for the CID-mediated organelle deformation.

### Optimization of PLAAT3 to minimize unfavorable characteristics: subcellular localization

PLAAT3 has two characteristics that could unwantedly complicate its use in functional assays. First, constitutive expression of PLAAT3 in cultured cells for 2 to 3 days leads to a decreased number of peroxisomes and loss of their internal contents[17,18,20], likely due to degraded peroxisomal membranes[17]. Second, constitutive expression of PLAAT3 in cancer cells induces cell death[27,28]. To mitigate the effect on peroxisome biogenesis and functions, we generated several PLAAT3 mutants specifically in the CT domain which binds to a peroxisomal biogenesis factor 19 (PEX19)[18]. We first fused YFP-FKBP to the C-terminus of full-length PLAAT3 (PLAAT3-FL-YFP-FKBP), intending to nullify the interaction between PLAAT3 and PEX19 with steric hindrance. Indeed, PLAAT3-FL-YFP-FKBP no longer localized to peroxisomes and became diffuse (Supplementary Fig. 2). We next truncated the CT domain (Fig. 1c), either partially (2CT) or fully (dCT), and observed its subcellular localization. We made truncations of the catalytically inactive LD version (C113S) specifically, in hopes to maintain the number of peroxisomes regardless of their localization. While mCherry-PLAAT3-FL-LD co-localized with peroxisomes, neither of its truncated counterparts, 2CT or dCT, was detected at peroxisomes (Supplementary Fig. 3a–c). Instead, the 2CT and dCT mutants were partially mislocalized to mitochondria and endoplasmic reticulum (ER), respectively (Supplementary Fig. 3d–i). The observed mitochondrial mislocalization of human 2CT was consistent with a murine version of 2CT described previously[20].

We then quantified the extent of rapamycin-triggered mitochondrial deformation with each of these PLAAT3 mutants, and found that FL, 2CT and dCT, but not FL-LD, worked effectively in inducing robust mitochondria deformation (Fig. 1d). Due to the mitochondrial pre-localization, 2CT deformed mitochondria even before rapamycin

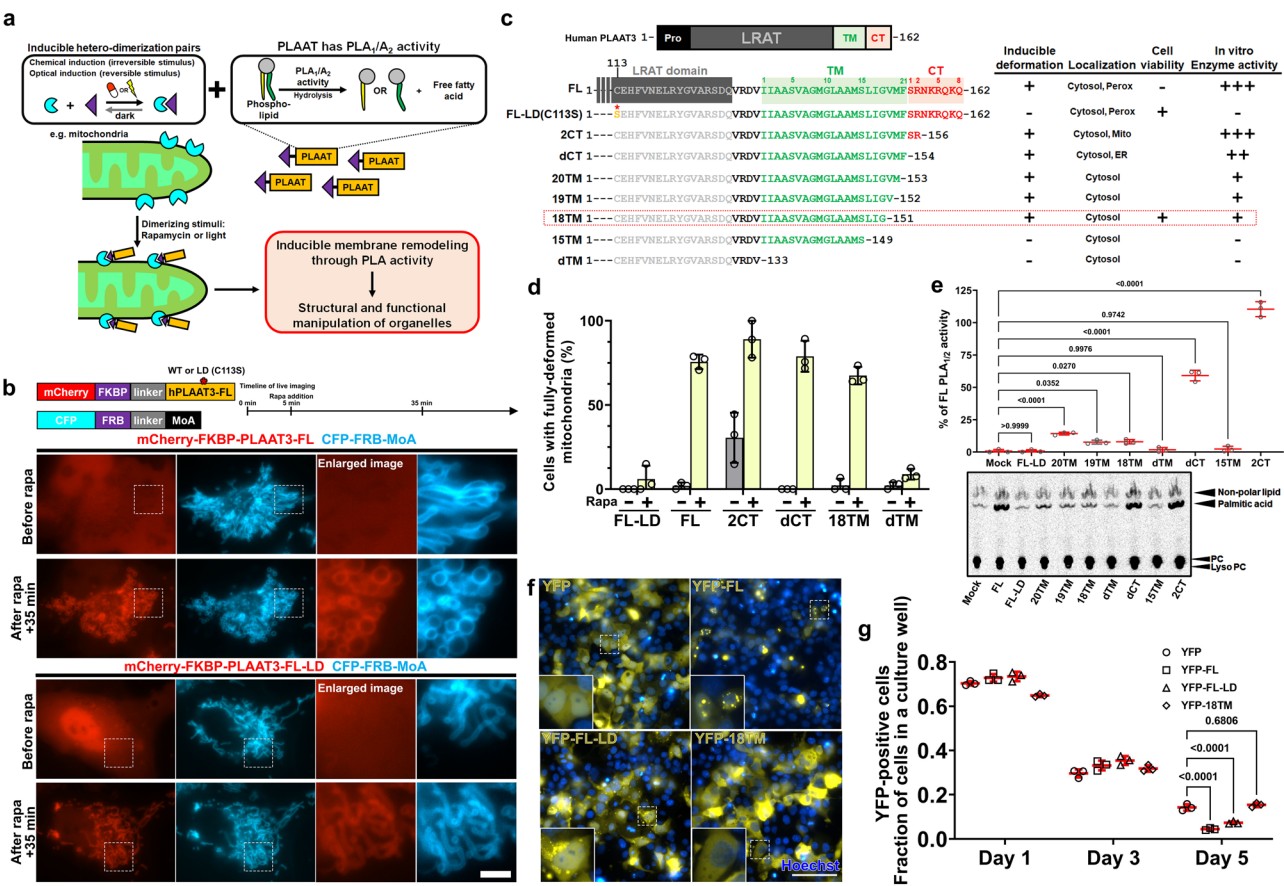

**Fig. 1 | Inducible mitochondrial manipulation by PLAAT3 and its optimization.**
**a** A conceptual schema describing rapidly inducible manipulation of intracellular organelles based on the integration of chemically or optically triggered hetero-dimerization and phospholipid remodeling by PLAAT enzymes. This strategy anticipates rapid switching of the PLAAT activity right at the target membrane-bound organelles (e.g., mitochondria) to induce their deformation as well as defunctionalization. **b** Fluorescence images of COS-7 cells expressing CFP-FRB-MoA along with either mCherry-FKBP-PLAAT3-FL or mCherry-FKBP-PLAAT3-FL-LD before and 35 min after addition of 100 nM rapamycin. Experimental timeline and schematic drawing of constructs are shown on top. The experiment was repeated more than three times. **c** A summary of PLAAT3 domain structures (top), as well as C-terminal sequences of full-length PLAAT3 along with its LD and a series of truncation mutants (bottom) and their properties. 18TM was used as a tool in the later experiments. Pro: proline-rich domain, LRAT: Lecithin-retinol acyltransferase domain, TM: putative transmembrane domain, CT: C-terminus domain, FL: full length of hPLAAT3, LD: lipase-dead

harboring C113S point mutation, 20TM, 19TM, 18TM, and 15TM: PLAAT3 mutant with truncations in TM and a full defect of CT, dTM: PLAAT3 without TM and CT. **d** The fraction of COS-7 cells indicating fully deformed mitochondria was calculated before and after rapamycin treatment, and presented for cells expressing mCherry-FKBP-FL-LD, mCherry-FKBP-FL or its mutants, and CFP-FRB-MoA. $n = 104, 110, 121, 96, 138, 115, 103, 102, 97, 93, 125,$ and $111$ cells from left to right; manually counted. **e** Cell homogenates (50 μg protein) were assayed for $PLA_{1/2}$ activity. The resultant radioactive products were separated by thin layer chromatography, and the activities ($n = 3$) were calculated by quantifying the produced palmitic acid using an image reader FLA-7000. **f** Fluorescence images of COS-7 cells expressing YFP, YFP-FL, YFP-FL-LD, or YFP-18TM, obtained at 48 h post-transfection. Scale bar, 100 μm. **g** The fraction of YFP-positive cells shown in **f** was calculated. $n = 591, 556, 499$ and $601$ cells from left to right. All data were analyzed from three individual experiments. Error bars, means ± s.d. Statistical significance was determined by two-way ANOVA with Dunnett's multiple comparison and $p$ values are indicated on the graph.

addition. Based on dCT mutant, we next designed a series of TM truncation mutants where TM amino acids were truncated (20TM, 19TM, 18TM, and 15TM) or in its entirety (dTM) (Fig. 1c). When we carried out the CID deformation assay using these mutants, 20TM, 19TM, and 18TM led to mitochondrial deformation comparable to FL, while the remaining mutants such as 15TM and dTM did not (Fig. 1c, d and Supplementary Fig. 4). To directly evaluate the $PLA_{1/2}$ activity of these PLAAT3 truncates, we performed an in vitro assay where FL and the truncation mutants were overexpressed in HEK293 cells. We first confirmed their expression in western blot analysis using half of the cell homogenates, which was later used to normalize enzymatic activity. We then incubated the remaining half of the homogenates with [$^{14}$C]dipalmitoyl-PC, and the generation of [$^{14}$C]palmitic acid as well as [$^{14}$C]lyso-PC were detected using thin layer chromatography. As a result, FL, 2CT, dCT, 20TM, 19TM, and 18TM indicated enzymatic activity, while FL-LD, 15TM and dTM did not (Fig. 1e). We determined that 20TM, 19TM, and 18TM retained 10-20% phospholipase activity of

the FL. Of note, the assay result could also reflect the affinity of tested proteins against the liposomes.

Considering 18TM as a promising candidate for the CID operation, we extensively examined its subcellular distribution and did not detect localization to any of the organelles tested (Supplementary Fig. 5a–j). To directly characterize the effect on peroxisome biogenesis, we quantified the number of peroxisomes using mScarlet-SRL (mSca-peroxi)[29] as a peroxisomal matrix marker. Accordingly, the number of mSca-peroxi-positive signals decreased in FL-expressing cells, but not in YFP- or YFP-18TM-expressing cells (Supplementary Fig. 6a, b). This result suggests that 18TM does not affect peroxisome biogenesis.

### Optimization of PLAAT3 to minimize unfavorable characteristics: cell viability
We next aimed to address the remaining unfavorable effect of PLAAT3, namely compromised cell viability which has been described in cancer cell lines[27,28]. We used a fluorescence microscope to count the cells

expressing either YFP, YFP-FL, YFP-FL-LD, or YFP-18TM after two days of transfection (Fig. 1f). The fraction of YFP-FL positive cells was significantly smaller (21.1%) than the cells expressing YFP (43.6%), YFP-FL-LD (37.5%) and YFP-18TM (47.4%). Since this experiment was prone to potentially heterogeneous transfection efficiency among these four constructs, we repeated the assay, but this time analyzing the fraction of YFP expressing cells using a flow cytometer at three different time points after transfection. As a result, all four constructs indicated similar protein expression on days 1 and 3, while cells expressing YFP-FL decreased significantly on day 5 (-70% reduction in relative to control cells expressing YFP), inferring the toxicity of the full-length PLAAT3. In contrast, cells expressing PLAAT3-18TM did not exhibit such decrease (Fig. 1g).

## Characterization of 18TM as a molecular tool

Membrane damage was reported to recruit PLAAT3 to organelle membranes[20]. We thus examined the extent of stress-induced recruitment of 18TM along with its full-length counterpart and found that 18TM does not exhibit this unfavorable property, likely due to the absence of CT and/or the incomplete TM (Supplementary Figs. 7, 8) (see Supplementary Note 1 for experimental details). In addition, we characterized the extent, kinetics and dynamics of mitochondria deformation by CID-induced 18TM recruitment (see Supplementary Note 2 for experimental details), where we observed deformation that developed over time with three discernible steps - blebbing, tearing, and shrinkage (Fig. 2 and Supplementary Movie 3). We further confirmed that the observed morphological change based on CFP-FRB-MoA fluorescence indeed resulted from actual morphological change of mitochondria with antibody staining against a mitochondria outer membrane protein, TOM20 (Supplementary Fig. 9).

To directly visualize mitochondrial phospholipid modification after the 18TM recruitment, we employed a fluorescence-conjugated phosphatidylserine (PS) where an NBD fluorescent dye is covalently attached to the Sn2 position. When added to culture media, this molecule localized to membranes of mitochondria. We reasoned that the 18TM recruitment leads to hydrolysis of PS, liberating an NBD-labeled fatty acid chain from the mitochondria membrane. By measuring NBD fluorescence at mitochondria before and after the 18TM recruitment, we could observe a significantly reduced NBD signal (Supplementary Fig. 10) (see Supplementary Note 3 for experimental details). All other control conditions did not lead to such NBD reduction.

Since DRP1 protein is indispensable for mitochondria fission[30,31], we asked if the 18TM-mediated mitochondria deformation such as the observed fragmentation is dependent on DRP1. To address this, we executed the CID operation using 18TM in WT and *Drp1* knockout mouse embryonic fibroblasts[32] (MEFs). Mitochondrial rounding and fragmentation were similarly observed in both cells (Supplementary Fig. 11a, b), indicating that the 18TM action is independent of DRP1.

Collectively, through the cycles of optimization and characterization summarized in Fig. 1c, we identified a PLAAT3 truncation mutant, namely 18TM, that is devoid of unfavorable characteristics originally associated with the full-length protein. Based on 18TM, we generated a molecular tool for synthetic induction of mitochondria deformation that depends on its phospholipase activity.

## Defunctionalization of mitochondria following chemically inducible translocation of 18TM

To test whether 18TM-mediated deformation has an influence on mitochondrial functions, we evaluated two representative properties: membrane potential which empowers ATP synthesis, and autophagic degradation of mitochondria known as mitophagy.

We first performed the CID-PLAAT3 experiments to deform mitochondria using YFP-FKBP-18TM (or YFP-FKBP-18TM-LD as a control) together with CFP-FRB-MoA in COS-7 cells. Tetramethyl rhodamine methyl ester (TMRE), an indicator for the mitochondria membrane potential, was then applied at 35 nM for 30 min prior to the onset of live-cell imaging. TMRE signals exhibited a moderate decrease 15–30 min after rapamycin addition, which continued till they became almost undetectable (Fig. 3a, b and Supplementary Movie 4). This signal decrease was not due to photobleaching or any systematic artifact, as the TMRE signal intensities were constant in 18TM-LD-transfected cells (Fig. 3b). To quantify this, we measured the TMRE fluorescence intensity at 30 and 90 min after rapamycin addition (Fig. 3c), which indicated that 18TM recruitment induced ~60% decrease in TMRE signal over the 60 min, while 18TM-LD recruitment did not exhibit such decrease. This suggests that mitochondria membrane potential was significantly lost after 18TM-induced mitochondria deformation. The membrane potential was dissipated even in the presence of 1 mM cyclosporin A, an inhibitor of mitochondrial permeability transition pore (mPTP) (Fig. 3d), suggesting that mPTP opening was not the major cause.

We next measured mitochondrial ATP using mitAT1.03, an organelle-targeted Förster resonance energy transfer (FRET) indicator of ATP[33]. Methods for measurement and analysis of CFP-to-YFP FRET were described previously[8,9,34]. Cells expressing mitAT1.03 were subjected to mitochondrial recruitment of 18TM for 2 h. As a result, the measured FRET values normalized by CFP intensity were significantly smaller than that with no recruitment of 18TM ($0.44 \pm 0.02$ and $0.37 \pm 0.01$, without and with 18TM recruitment, respectively, $p = 0.0027$) (Fig. 3e, f, left). This difference was not observed when 18TM-LD was used instead of 18TM ($0.50 \pm 0.03$ and $0.48 \pm 0.06$, without and with 18TM-LD recruitment, respectively, $p = 0.88$) (Fig. 3e, f, right). As a control, 25 mM of 2-deoxy-D-glucose was used to lower ATP level, and indeed we observed a striking decrease in FRET signal. This data suggests that 18TM-induced mitochondria deformation leads to reduction in ATP synthesis likely due to the waning membrane potential.

We then examined if mitophagy took place following the 18TM-mediated mitochondria deformation. Mitophagy was monitored by accumulation of YFP-Parkin at mitochondria[8]. Cells transfected with mCherry-FKBP-18TM (or mCherry-FKBP-18TM-LD as a control), CFP-FRB-MoA and YFP-Parkin were subjected to 6 h rapamycin treatment, and the fraction of cells indicating Parkin localization at mitochondria was calculated (Fig. 3g, h). Accordingly, we did not observe any sign of mitophagy under 18TM or 18TM-LD conditions, regardless of the striking deformation of mitochondria induced by 18TM. We used carbonyl cyanide 3-chlorophenylhydrazone (CCCP) as a positive control, and as expected, a large fraction of cells (63.4%) accumulated YFP-Parkin at the mitochondria (Fig. 3g, h). We also analyzed colocalization between deformed mitochondria and LC3 or lysosomes (Fig. 3i, j). As a result, we did not observe colocalization of deformed mitochondria with LC3 or lysosomes (LC3: $-0.19 \pm 0.03$, lysosomes: $-0.20 \pm 0.16$). These values were not significantly different from those obtained with mitochondria of normal morphology in 18TM-LD transfected cells: LC3 ($-0.08 \pm 0.19$, $p = 0.43$) and lysosomes ($-0.18 \pm 0.14$, $p = 0.87$), suggesting that 18TM-mediated deformed mitochondria were unlikely to be undergoing mitophagy at least within 6 h of the deformation. Collectively, these data suggest that 18TM triggered loss of membrane potential, decrease in ATP synthesis, but not Parkin-dependent mitophagy.

## Loss of luminal proteins and ruptures of outer membranes

To understand how the remodeling of phospholipids and subsequent deformation led to a loss of the electric potential of mitochondria membranes and ATP synthesis, we examined possible leakage of a mitochondria-resident soluble protein, Subunit 9 of mitochondrial ATPase (Su9) in the matrix[35]. When YFP-FKBP-18TM (or YFP-FKBP-18TM-LD as a control), Su9-CFP and mCherry-FRB-MoA were co-transfected in COS-7 cells, we observed a decrease of Su9-CFP signals

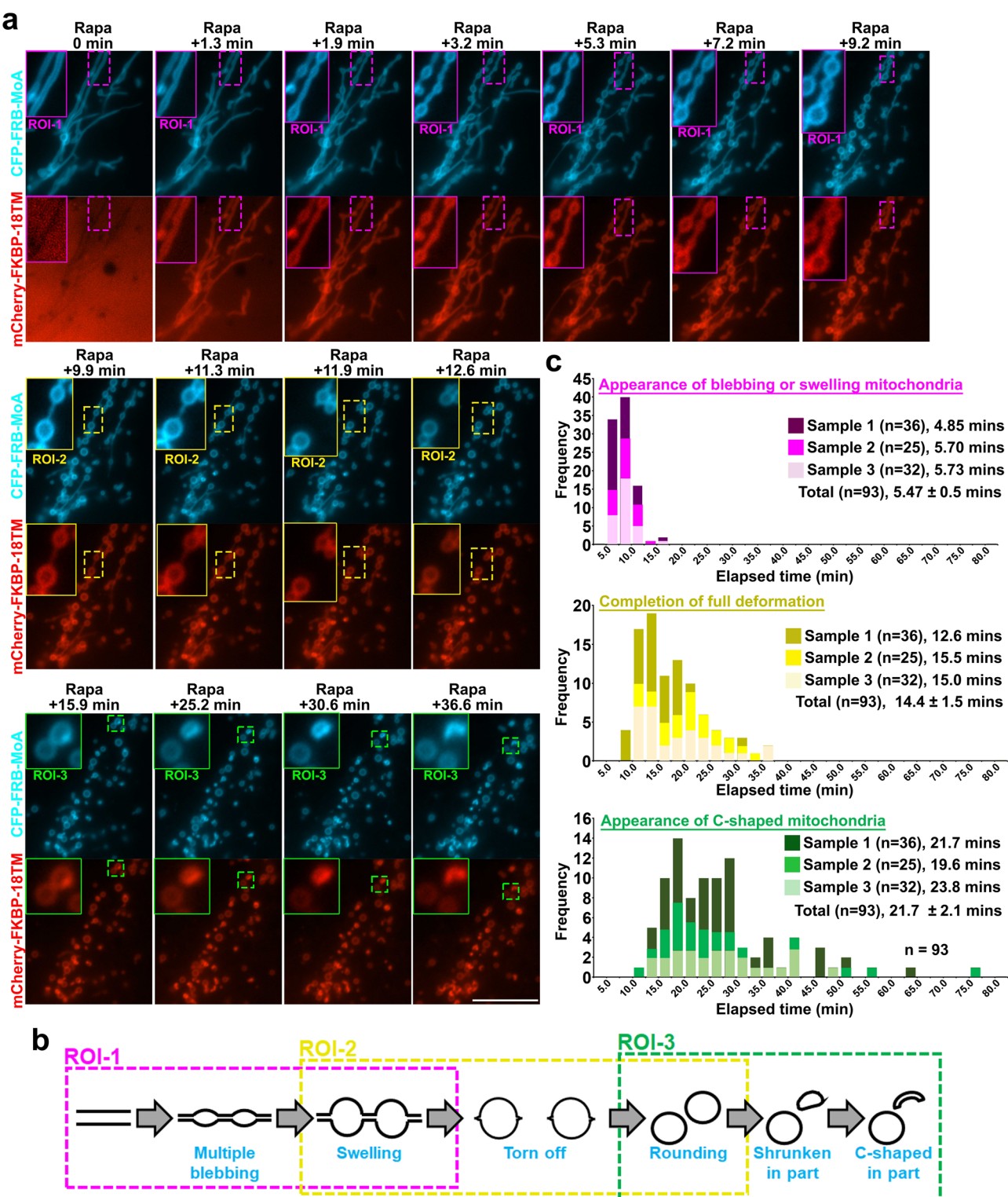

**Fig. 2 | Kinetics of mitochondria morphological changes following induction of 18TM translocation. a** Fluorescence images of cells expressing CFP-FRB-MoA (cyan) and mCherry-FKBP-18TM (red), indicating three-step morphological changes. ROI-1: Straight-form mitochondria changed into multiple blebbed structures (top panels). ROI-2: Blebs further swelled and finally torn off (middle panels). ROI-3: A part of rounding mitochondria changed into shrunken or C-shaped form (bottom panels). Scale bar = 10 μm. The experiment was repeated at least three times. **b** Schematic illustration of the three-step deformation corresponding to the images in **a**. **c** The number of cells with the indicated deformation type (top: blebbing/swelling, middle: tearing/rounding, bottom: shrinkage/C-shaping) was counted at each time point and presented as a histogram. Ninety-three cells in total were manually counted from three different experiments (36 cells for sample 1, 25 cells for sample 2 and 32 cells for sample 3). The median value from each sample as well as mean ± s.d. are indicated in a histogram which was generated by Microsoft Excel. Rapa: rapamycin. ROI: region of interest.

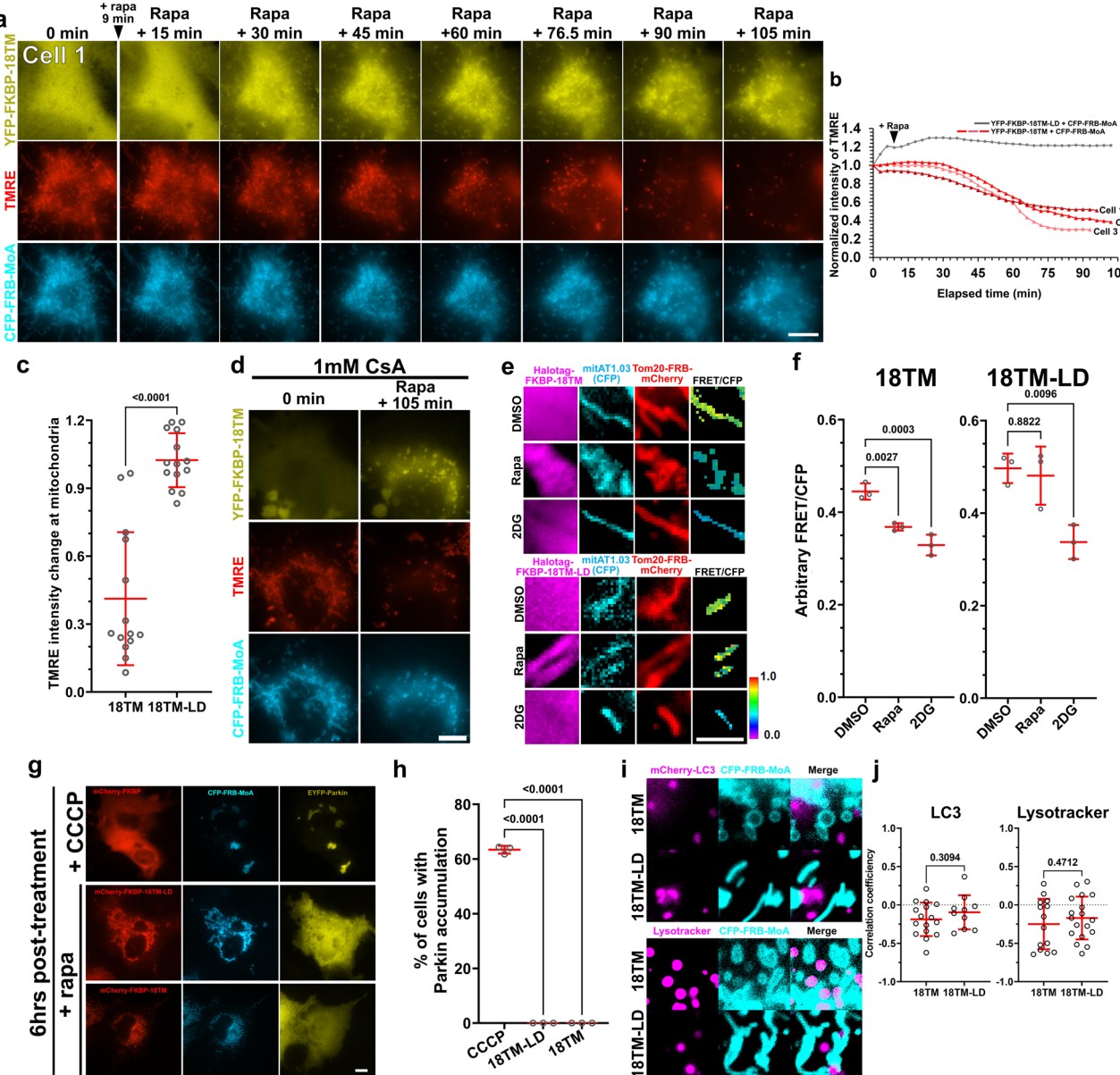

**Fig. 3 | Effect of PLAAT3-18TM-mediated deformation on functional aspects of mitochondria. a** Time-lapse fluorescence images of TMRE in cells expressing YFP-FKBP-18TM (or -18TM-LD) and CFP-FRB-MoA before and after rapamycin treatment. **b** The fluorescence intensity of TMRE in three different cells was plotted as a function of time after normalized to the value at *t* = 0. **c** The TMRE fluorescence intensity in cells with recruitment of 18TM or 18TM-LD was measured at 30 and 90 min after rapamycin addition, and indicated as a ratio from three independent experiments. **d** Fluorescence images of YFP-FKBP-18TM, TMRE and CFP-FRB-MoA in the presence of 1 mM CsA (added for 1 h prior to the imaging). **e, f** Mitochondrial ATP measurement with or without 18TM-induced mitochondria deformation. Representative FRET images of mitAT1.03 along with TOM20-mCherry-FRB and either HaloTag-FKBP-18TM (left panels) or HaloTag-FKBP-18TM-LD (right panels). FRET images are shown as a ratio against CFP intensity of mitAT1.03 in pseudo-colors, while the Halo Tag constructs are visualized by JF646 dye-labeled Halo Tag ligand. Cells were treated for 2 h with one of the three drugs: 0.1% DMSO (top), 100 nM rapamycin (middle), 25 mM 2-DG (bottom) (**e**). Scale bar, 5 μm. Arbitrary FRET/CFP ratio values were obtained from three independent experiments (*n* = 3 biologically independent cells) (**f**). **g** Fluorescence images of COS-7 cells expressing YFP-Parkin and CFP-FRB-MoA along with either mCherry-FKBP, -18TM or -18TM-LD. Cells were treated with 10 μM CCCP or rapamycin for 6 h. **h** A fraction of cells indicating co-localization of YFP-Parkin and mitochondria was calculated (manually counted). *n* = 168, 294 and 241 cells from left to right; analyzed from three individual experiments. **i, j** Based on the fluorescence images of cells co-expressing CFP-FRB-MoA and either mCherry-FKBP-18TM or -18TM-LD, and either expressing mCherry-LC3 or stained with Lysotracker, which were treated with rapamycin for 6 h (**i**), their colocalization coefficients were measured (*n* = 16, 10, 14, 18 from left to right, biologically independent cells from three independent experiments) (**j**). Error bars, means ± s.d. Statistical significance was determined by one-way ANOVA with Dunnett's multiple comparison and *p* values are indicated on the graph. Scale bar, 10 μm. Rapa: rapamycin.

upon 18TM-induced deformation (Fig. 4a), which appeared to be discrete at individual mitochondrion (Fig. 4b and Supplementary Movie 5). None of these were observed upon recruitment of 18TM-LD (Fig. 4c and Supplementary Movie 6). To quantify the observed changes, we measured fluorescence intensity of CFP-Su9 at mitochondria at 30 and 120 min after recruitment of 18TM/18TM-LD. The

calculated ratio intensity indicates modest decrease over the 90 min of 18TM recruitment, which was significantly different from the corresponding value obtained with 18TM-LD (0.92 ± 0.10 vs. 1.09 ± 0.12, *p* < 0.001) (Fig. 4d). These results made us think that the recruitment of 18TM could increase permeability of the mitochondrial membranes. Though the leakage of Su9-CFP signals to the cytosol may potentially

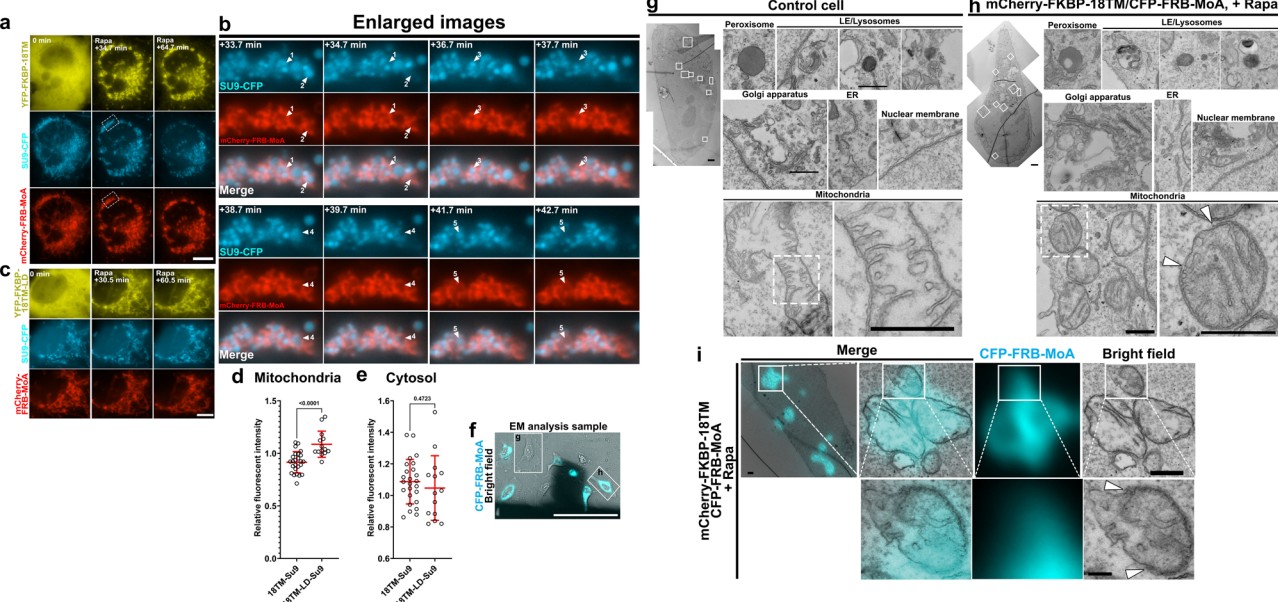

**Fig. 4 | Leakage of luminal proteins following 18TM-mediated mitochondrial deformation. a–c** Time-lapse fluorescence images of mCherry-FRB-MoA and Su9-CFP (a mitochondria matrix marker), along with either YFP-FKBP-18TM (**a**) or YFP-FKBP-18TM-LD (**b**), in COS-7 cells before and after treatment with 100 nM rapamycin. Images indicated by dotted white squares in **a** are magnified in **b**. The experiments were repeated at least three times. Scale bar, 10 μm. **d** Fluorescence intensity of CFP-Su9 at the deformed mitochondria at 30 min after recruitment of 18TM/18TM-LD divided by the intensity at 120 min. We intentionally avoided characterizing the first 30 min during which mitochondria undergo striking morphological changes, preventing accurate measurement of the fluorescence signals. The calculated ratio intensity indicates modest decrease over the 90 min of 18TM recruitment, which was significantly different from the corresponding value obtained with 18TM-LD. **e** Fluorescence intensity of CFP-Su9 in the cytosol was quantified at 30 and 120 min after recruitment of 18TM/18TM-LD. The calculated ratio intensity indicates that there is no significant increase upon 18TM recruitment nor difference between 18TM and 18TM-LD. The signal intensities were measured

with NIS element software. Error bars, means ± s.d. **f–h** Ultrastructural analysis of organelles following 18TM-mediated mitochondrial deformation. HeLa cells transfected with CFP-FRB-MoA and mCherry-FKBP-18TM (**f**). Merged bright field and fluorescence images of un-transfected (**g**) and transfected (**h**) cells 30 min after the addition of rapamycin. Scale bar, 100 μm. Representative transmission electron microscopy (TEM) images of organelles in un-transfected control (**g**) and transfected cells (**h**). Organelles in the white boxes (solid line) in the left panel are enlarged on the right to show the integrity of the annotated organelles. For mitochondria, the region within the white box (dotted line) is zoomed in further to show the integrity of outer and inner mitochondrial membrane. Arrowheads indicate the edges of ruptured outer mitochondrial membranes. Scale bar, 2 μm (whole cell) and 500 nm (organelles). **i** Correlative EM analysis of mitochondria in the 18TM/MoA-transfected cell. Arrowheads indicate the edges of the ruptured outer mitochondrial membrane. The experiment was performed once (**f–h**). Scale bars, 2 μm (left low magnified panel), 500 nm (top panels), and 100 nm (bottom panels). Rapa: rapamycin.

increase the cytosolic fluorescent levels, no such increase was observed in 18TM- or 18TM-LD-expressing cells (Fig. 4e).

Next, we examined membrane integrity of the deformed mitochondria using correlative electron microscopy (EM). Cells transfected with YFP-FKBP-18TM and CFP-FRB-MoA and treated with rapamycin for 30 min were subjected to fluorescence imaging (Fig. 4f), followed by EM analysis (Fig. 4g, h). Un-transfected cells serving as a negative control in the same culture featured a normal tubular form of mitochondria, and membrane integrity and cristae morphology also appeared normal (Fig. 4g). In contrast, deformed mitochondria in 18TM-transfected cells exhibited rupture of the mitochondrial outer membrane (Fig. 4h). The ruptured membrane did not seem to overlap well with the CFP-FRB-MoA fluorescence signal (Fig. 4i).

Of importance, other membrane-bound organelles, such as peroxisomes, lysosomes, the Golgi apparatus, endoplasmic reticulum, and the nuclear envelope, did not show abnormalities in membrane integrity or overall morphology (Fig. 4g, h). We also validated the organelle specificity in live-cell fluorescence imaging where induction of mitochondrial defunctionalization did not affect peroxisomal properties such as retention of luminal proteins (Supplementary Fig. 12a, b). These results support the high organelle specificity of the 18TM-mediated defunctionalization, at least for ~30 min. As organelles are known to interact with each other and exchange materials, it is possible that defunctionalization of a given organelle impacts other organelles after some time.

## Optogenetic deformation of mitochondria

To explore the utility of our PLAAT strategy for organelle defunctionalization, we next aimed to achieve an optogenetic operation with 18TM. In particular, we replaced the FKBP-FRB dimerizing pair with an mSspB-iLID pair, which dimerizes upon blue-light exposure. We generated two constructs: 18TM fused to a Halo Tag and mSspB (HaloTag-mSspB-18TM) and mitochondria-anchored iLID (iLID-MoA), which were co-transfected in cells along with TOM20-mCherry in trans to visualize mitochondria. Transfected cells were exposed to intermittent irradiation with blue light (447 nm) for 200 ms at a frequency of 1 min. Within minutes of the light exposure, we observed that recruitment of HaloTag-mSspB-18TM to the mitochondria (Fig. 5a) and quantified the fraction of cells with fully deformed mitochondria in fields of view (*n* = 50 for 18TM-transfected cells and *n* = 47 for 18TM-LD-transfected cells from three independent experiments). Optogenetic recruitment of 18TM resulted in deformation such as fragmentation and swelling in 51% of transfected cells (Fig. 5c). As a control, we repeated the optogenetic operation with HaloTag-mSspB with 18TM-LD and found little to no mitochondrial deformation (Fig. 5b). No cells with fully deformed mitochondria were observed (Fig. 5c).

## PLAAT-18TM-mediated mitochondria deformation in primary neuronal cultures via viral transduction

We thus far demonstrated the 18TM-mediated mitochondria defunctionalization in cultured cells. Molecular tools like this are useful if

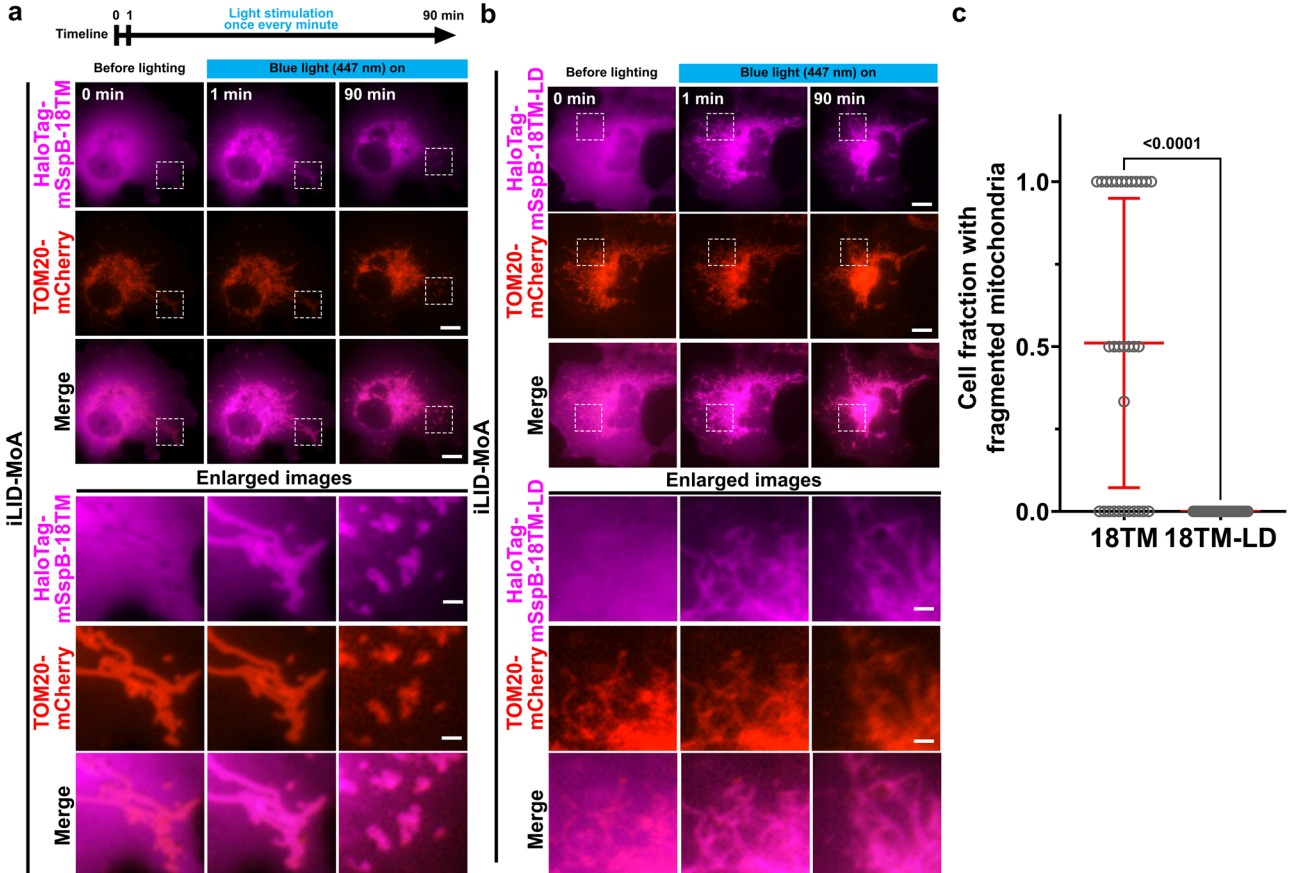

**Fig. 5 | Optogenetic control of 18TM-mediated mitochondrial deformation. a, b**
Time-lapse fluorescence images of COS-7 cells transfected with HaloTag-mSspB-18TM (**a**) or HaloTag-mSspB (**b**) along with iLID-MoA (a mitochondrial anchor) and mCherry-MoA (an indicator of mitochondria morphology). The cells were illuminated by blue light at 447 nm (blue bars on top). The HaloTag-mSspB was visualized with a JF646-conjugated HaloTag ligand which was incubated for 30 min before onset of the imaging. Areas marked by dashed-line boxes are magnified at the bottom of **a**. Live-cell imaging was performed in indicated timeline. The experiment was repeated more than three times. Scale bar = 10 µm. **c** A fraction of cells that indicated fragmentation and/or swelling was quantified for 18TM- and 18TM-LD-mediated optogenetic operations. [means ± s.d., respectively ($n = 31$ for 18TM and $n = 28$ for 18TM-LD; analyzed from three individual experiments)]. Cells with fully deformed mitochondria were manually counted. Statistical significance was determined by unpaired two-tailed $t$-test and $p$ values are indicated on the graph (**c**).

applicable to more physiologically relevant samples. We therefore delivered YFP-FKBP-18TM and TOM20-CFP-FRB by adeno-associated virus (AAV) vector into mouse primary hippocampal neurons. More specifically, we raised two types of AAV constructs: YFP-FKBP-18TM (or YFP-FKBP as a control) and TOM20-CFP-FRB, both under the control of a CMV promoter (Supplementary Fig. 13a). TOM20 is a component of translocase of the outer membrane (TOM) complex and its N-terminal signal sequence has been interchangeably used to target a protein of interest to mitochondria[22]. Hippocampal neurons were dissected from embryonic mice (E18), then cells were seeded and cultured for 6 days to sufficiently project neurites. At 6 days in vitro (DIV6), neurons were infected with two AAVs (YFP-FKBP-18TM and TOM20-CFP-FRB) at multiplicity of infection (MOI) = 40,000 and incubated for 2 days. Subsequent rapamycin treatment triggered deformation of mitochondria, such as dilation, fragmentation and swelling (Supplementary Fig. 13b), which was not seen in negative control samples where YFP-FKBP was introduced. These experiments demonstrate applicability of the present PLAAT strategy to different sample preparations beyond culture cell lines.

**Rapidly Inducible Defunctionalization of Peroxisomes**
To test if the PLAAT strategy can be generalized to other membrane-bound organelles, we chose peroxisomes, since unlike mitochondria there is only a limited number of techniques available for perturbation of peroxisome functions[36]. To direct 18TM to the cytosolic surface of

peroxisomes, we used peroxisomal membrane protein PEX3[37]. PEX3 fused to CFP-FRB (PEX3-CFP-FRB) showed high colocalization with a peroxisomal protein such as catalases (correlation coefficient: 0.85 ± 0.01, 60 cells in total from three independent experiments). PEX3-CFP-FRB was then transfected in HeLa cells along with YFP-FKBP-18TM (or 18TM-LD as a control) and the matrix marker mSca-Peroxi. Rapamycin addition successfully recruited PLAAT3-18TM to peroxisomes on a timescale of minutes (Fig. 6a, b, Supplementary Movie 7) which indicated that mSca-Peroxi signals were lost sharply and discretely after 18TM recruitment, while YFP-18TM and PEX3-CFP-FRB remained at the organelle. This feature was not observed in YFP-18TM-LD-expressing control cells (Fig. 6c). To quantify these observations, we manually counted the number of mSca-Peroxi positive puncta before and after rapamycin addition, and confirmed the striking decrease with YFP-18TM, but not with 18TM-LD (Fig. 6d and Supplementary Movie 8). If the loss of peroxisomal luminal proteins was due to their leakage rather than other mechanisms such as protein degradation, one may expect concomitant increase of mSca-peroxi signals in the cytosol. Therefore, we measured fluorescence intensity of mSca-peroxi in the cytosol, which indeed exhibited an increase in 18TM-expressing cells, but not in 18TM-LD-expressing cells (Fig. 6e).

To test if endogenous matrix proteins also leaked out of peroxisomes, we next conducted immunocytochemistry in cells expressing YFP-FKBP-18TM (or 18TM-LD) and PEX3-CFP-FRB using an antibody against a catalase, an enzyme which decomposes $H_2O_2$. In 18TM-

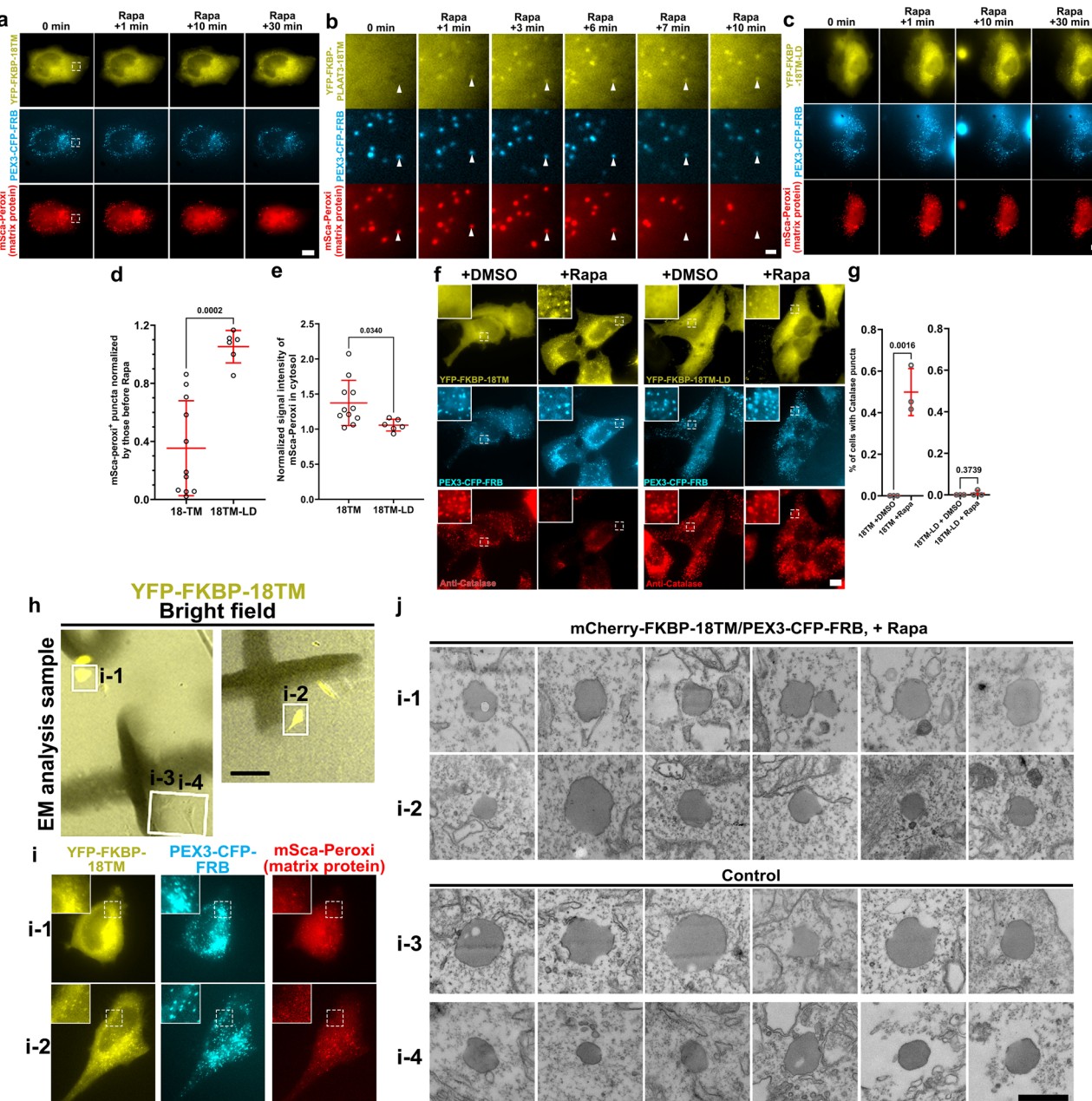

**Fig. 6 | Rapid peroxisome defunctionalization by a chemically inducible PLAAT3-18TM. a–c** Time-lapse fluorescence images of peroxisomal luminal proteins (mSca-Peroxi) in HeLa cells are presented upon addition of 100 nM rapamycin to trigger chemically induced recruitment of PLAAT3-18TM (**a**) or PLAAT3-18TM-LD (**b**). Images indicated by dotted white squares are magnified in **c**. **d** Quantification of the fraction of mSca-peroxi⁺ puncta ($n = 11$ for 18TM and $n = 6$; analyzed from three individual experiments). Signal intensities were measured with NIS element software. **e** Quantification of normalized signal intensities of mSca-Peroxi from cytosol in 18TM- and 18TM-LD-transfected cells ($n = 10$ for 18TM and $n = 6$; analyzed from three individual experiments). Data in **d** and **e** were normalized by values before rapamycin treatment. mSca-peroxi puncta were manually counted. **f** The transfected HeLa cells with indicated constructs were treated with either DMSO or 100 nM rapamycin for 2 h before chemical fixation, followed by immunostaining against endogenous catalases. **g** The fraction of cells without catalase signals was calculated based on the

fluorescence images in **f**, where dots indicate individual data points and means ± s.d., respectively ($n = 112$, 121, 111, and 119 cells from left to right; analyzed from three individual experiments). Cells with or without Catalase puncta were manually counted. Rapa: rapamycin. Scale bar = 1 μm for **b** and 10 μm for **a**, **c**, and **f**. **h**–**j** Ultrastructural analysis of peroxisomes following 18TM-mediated perturbation of peroxisomes. **h** HeLa cells were transfected with YFP-FKBP-18TM (yellow), PEX3-CFP-FRB (cyan) and mSca-Peroxi (red). Merged bright field and fluorescence images of transfected cells (i-1 and -2) and un-transfected control cells (i-3 and -4) 30 min after the addition of rapamycin are shown in left panels. **i** The leakage of mSca-Peroxi was observed in the transfected cells (i-1 and -2, right panels). Scale bar = 100 μm (left panels) and 10 μm (right panels). **j** TEM images of peroxisomes in the PEX3/18TM-transfected cells (i-1, i-2) and control cells (i-3 and i-4). Scale bar = 500 nm. Statistical significance was determined by unpaired two-tailed *t*-test (**d**, **e**) and one-way ANOVA with Dunnett's multiple comparison (**g**). *p* values are indicated on graphs.

expressing cells treated with a vehicle (DMSO), catalase was confirmed to localize at peroxisomes (Fig. 6f). However, rapamycin treatment in the same cells led to significant decrease in catalase staining (Fig. 6f, g). This dissipation of catalase punctate signals did not occur in 18-TM-LD-

expressing cells (Fig. 6f, g). Catalase-mediated ROS decomposition inside the matrix represents one of the critical functions of peroxisomes[38]. Indeed, mislocalization of peroxisomal catalases to the cytosol often associates with defective *Peroxin* genes that are

responsible for peroxisomal biogenesis[36,37]. Since the 18TM operation led to the loss of such catalases, this particular peroxisomal function is likely impaired. For an ultrastructural analysis of the peroxisomal membranes, we obtained EM images of peroxisomes in cells undergoing a loss of the peroxisomal mSca-peroxi signal upon rapamycin-induced 18TM recruitment (Fig. 6h, i). As a result, there are no apparent ultrastructural features in these cells that are different from those in un-transfected neighboring cells (Fig. 6j).

Lastly, we examined the organelle specificity by performing live-cell fluorescence imaging, where induction of peroxisomal defunctionalization did not impact mitochondria morphology (Supplementary Fig. 14).

## Discussion

PLAAT members possess phospholipase A/acyltransferase activity along with promiscuous substrate specificity. With these unique characteristics, PLAAT3 is associated with the degradation of almost all types of organelles in eye fiber cells for lens clearing[20]. Despite the strong potential, PLAAT3 has features that impeded its readily implementation for synthetic operations in living cells. Consistent with previous reports[17-19,27,28], full-length PLAAT3 localized to peroxisomes (Supplementary Fig. 3), impaired their biogenesis (Supplementary Fig. 6), and, when overexpressed, compromised cell viability (Fig. 1). In addition, we found that environmental factors such as oxidative and hyperosmotic stress can trigger recruitment of PLAAT3 to mitochondria (Supplementary Figs. 7, 8). Combining live-cell fluorescence imaging, correlative EM analysis and in vitro characterizations, we identified the 18TM truncation mutant to retain the desirable membrane deformation ability, albeit without the abovementioned undesirable features. Chemically induced recruitment of 18TM to mitochondria led to acute loss of luminal proteins, membrane potential, and ATP synthesis, thus successfully defunctionalizing the organelle in a manner dependent on the PLA activity.

One may wonder how the phospholipase A/acyltransferase action by PLAAT on the outer membranes of mitochondria was translated into the leakage of mitochondrial inner proteins such as Su9. There is a protein complex, Ups1-Mdm35, in the mitochondrial intermembrane space that can transfer phospholipids between the two mitochondrial membranes[39,40]. We thus speculated that the lysophospholipids produced at the outer membranes could be routed to the inner membranes, while phospholipids of inner membranes could be simultaneously processed by PLAAT after being transported to the outer membranes.

In the process of PLAAT3 optimization, we found that a full-length PLAAT3 accumulates at the mitochondria in response to oxidative and hyperosmotic stresses (Supplementary Figs. 7, 8). Understanding physiological cues that trigger PLAAT3 recruitment to organelle membranes is important[20]. An oxidative insult such as $H_2O_2$ administration leads to peroxidation of membrane phospholipids, which in turn changes the membrane shape and increases the permeability, likely through bursting or pore formation[41,42]. If this applies to mitochondria, the $H_2O_2$ exposure may have caused physical damage to the outer membranes and subsequently induced recruitment of full-length PLAAT3. Of note, hyperosmotic shock can stimulate ROS production in corneal epithelial cells[43], thus could also induce membrane damage. Hyperosmotic shock may have an alternative or additional role - molecular crowding. Lens fiber cells undergo molecular crowding due to the concentration of crystallin proteins to attain the optimal refractive property[44-47]. Our data implied that membrane damage and/or molecular crowding could be a physiological trigger of PLAAT3 recruitment to organelles in lens.

The extent of the organelle defunctionalization should ideally be tunable. Besides varying the rapamycin concentration along with modifying expression levels of dimerizing fusion proteins (FKBP-18TM/FRB-MoA, etc.), PLAAT family members other than PLAAT3, or even

other members of the PLA superfamily, from diverse organisms ranging from virus and bacteria to mammals[10], could be adapted and engineered for this purpose. We expect such expansion to be made relatively feasibly thanks to the modular design of our molecular tool (i.e., simply replacing 18TM with other $PLA_1/A_2$ proteins). Since PLAAT1, 2, and 4 indicated striking mitochondrial deformation (Supplementary Fig. 1) and since these members have varying substrate specificity and catalytic strength[10], it is of our interest to develop defunctionalization tools based on these PLA members and to see if they exhibit different characteristics. An LRAT protein that catalyzes vitamin A esterification is another valid candidate, as this enzyme has a highly similar sequence and structure to the PLAAT family[48,49]. Other exciting possibilities include multiplexing of the present PLAAT tools and existing molecular tools for membrane deformation such as rupture[3-5] and bending[8,50], where their co-operation may synergize each other's action.

Notably, our PLAAT tool is an addition to a relatively limited pallet of peroxisome research tools. So far, gene modification of *Peroxin* genes and treatment with a catalase inhibitor such as 3-Amino-1,2,4-triazole have been primarily used to investigate physiological roles of this organelle[51-54]. Since our molecular tool is genetically encoded, its application should be scalable from single cells to tissues and animals, provided that the introduction of a dimerizer molecule or illumination of blue light can be feasibly achieved in these samples. As knockout of *Peroxin* genes in mice associates with deleterious phenotypes including embryonic fatality[55], it has been challenging to address the physiological role of peroxisomes in mice after development unless conducting a laborious conditional knockout. The present PLAAT-based synthetic operation to induce acute peroxisome defunctionalization in adult mice could thus become as powerful as the conditional knockout, albeit without preparing generations of genetically engineered mice. Once again, the modular design principle should enable the expansion of our PLAAT tools beyond mitochondria and peroxisomes. With a collection of these molecular tools, a new type of approaches to organelle biology will become possible.

## Methods

### Ethical statement

The present research complies with all relevant ethical regulations of Biosafety Office Health, Safety & Environment at Johns Hopkins University for human cell lines, recombinant DNAs, and viruses. All mouse neuron experiments were performed according to the National Institutes of Health with approval from the Animal Care and Use Committees at the Johns Hopkins University School of Medicine.

### Plasmid construction

If not otherwise specified, all plasmids expressing FKBP or FRB were constructed by standard subcloning techniques based on either *pECFP*, *pEYFP* and *pmCherry* plasmids (Clonetech). Human *Plaat* cDNAs were purchased from ORIGENE [RC208444 for *PLAAT1* (NM_020386), RC212578 for *PLAAT2* (NM_017878), RC200242 for *PLAAT3* (NM_007069), RC201923 for *PLAAT4* (NM_004585), and RC228184 for *PLAAT5* (NM_054108)]. For construction of *pAAV-CMV-YFP-FKBP-18TM*, *pAAV-CMV-YFP-FKBP-18TM-LD*, and *pAAV-CMV-TOM20-CFP-FRB*, *pAAV-MCS* (Cell Biolabs) was digested with EcoRI and BamHI, and fragments of *YFP-FKBP-18TM*, *YFP-FKBP* and *TOM20-CFP-FRB* were inserted to the digested sites using In-Fusion Snap Assembly Master Mix (Takara Bio USA, 638947). To visualize Golgi, lysosomes, autophagosomes, endosomes and nuclear membrane, *Giantin S*, *Lamp1*, and *Su9* cDNA were inserted to *pECFP* and cDNAs of *Lc3*, *Rab5*, and *Lamin A* were inserted to *pmCherry* by standard subcloning technique. *Parkin* cDNA was also subcloned to *pEYFP*. For PEX3-CFP-FRB, cDNAs encoding human *PEX3*, *mCerulean3* and *FRB* were inserted into *pcDNA3.1(+)* (Invitrogen) vector by standard restriction enzyme subcloning. *mScarlet-SRL*[29] was obtained from Addgene (#85065). All plasmids were verified by Sanger sequencing.

## Cell culture

HeLa cells, COS-7 cells, HEK293T cells (ATCC) and MEFs were maintained in DMEM (Corning, 10-013-CV) supplemented with 10% FBS (Sigma Aldrich, F6178), 1% Pen-Strep (Thermo Fisher, 15140163) at 37 °C under 5% CO$_2$. These cell lines have not been tested routinely for *Mycoplasma* contamination. In all experiments of chemical dimerization, rapamycin was treated at a final concentration of 100 nM.

## Alignment of amino acid sequences of PLAAT family proteins

Amino acid sequences of human PLAAT family proteins (NP_065119 for PLAAT1, NP_060348 for PLAAT2, NP_009000 for PLAAT3, NP_004576 for PLAAT4, and NP_473449 for PLAAT5) were obtained from the NCBI database and aligned by Clustal Omega.

## Cell viability assay

COS-7 cells were transfected with plasmids expressing YFP-FKBP, YFP-FKBP-FL, YFP-FL-LD, YFP-FKBP-18TM. Cells were incubated for 48 h and stained with Hoechst33342 (Invitrogen, H3570). Cell pictures were taken using an Eclipse Ti inverted fluorescence microscope (Nikon) equipped with ×20 objective lens (Nikon). YFP-positive cells and YFP-negative cells were manually counted using pictures taken from three different visual fields for each sample and the proportion of YFP-positive cells was calculated as cell viability. Per a visual field, 103–237 cells from three different experiments were counted.

For the flow cytometric analysis, HeLa cells were transfected with plasmids expressing YFP, YFP-FL, YFP-FL-LD, and YFP-18TM using FuGENE HD according to the manufacturer's protocol. Briefly, 1×10^5 HeLa cells were plated in 12-well plates a day before the transfection and transfected with 1 μg of plasmid DNA and 3 μl of FuGENE HD per well. Then all the cells in each well were collected at day3 and day5 after transfection and analyzed with Spectral Cell Analyzer SA3800 (Sony). The YFP-positive populations in living cells were quantified to see the toxic effects by PLAAT3 and PLAAT3 mutants.

## Enzyme activity assay

HEK293 cells were transfected with the expression vector harboring PLAAT3 or its mutants by the use of Lipofectamine 2000 according to the manufacturer's instructions. Two days after transfection, the cells were harvested and sonicated in homogenization buffer (50 mM Tris-HCl, pH 7.4) three times each for 3 s. To measure PLA$_{1/2}$ activity, the cell homogenates (50 μg protein) were incubated with 200 μM 1,2-[1'-$^{14}$C]dipalmitoyl-PC (45,000 cpm) in 100 μl of 50 mM Tris-HCl (pH 8.0), 2 mM DTT and 0.1% Nonidet P-40 at 37 °C for 30 min. The reaction was terminated with the addition of 320 μl of a mixture of chloroform/methanol (2:1, v/v) containing 5 mM 3(2)-*t*-butyl-4-hydroxyanisole. After centrifugation, 100 μl of the lower fraction was spotted on a silica gel thin-layer plate (10-cm height) and developed at 4 °C for 25 min in chloroform/methanol/H$_2$O (65:25:4, v/v). The distribution of radioactivity on the plate was visualized and quantified using an image reader FLA-7000 (FUJI-FILM, Tokyo, Japan).

## Visualization of fluorescently labeled phosphatidylserine

A fatty acid-labeled phosphatidylserine (1-palmitoyl-2-{12-[(7-nitro-2-1,3-benzoxadiazol-4-yl)amino]lauroyl}-sn-glycero-3-phosphoserine, or 16:0-12:0 NBD-PS, Avanti Polar Lipids, 810193) in Chloroform was dried up under nitrogen gas, resolved in ethanol, and then added to phosphate-buffered saline containing defatted BSA (Sigma, A3803) with a rigorous vortex. The NBD-PS solution was then added to HeLa cells with an incubation of 30 min at 37 °C, prior to fluorescence imaging under epi-fluorescence microscope. A CFP channel was used to capture the NBD fluorescence signal. Signal intensities were measured with MetaMorph imaging software.

## AAV production and infection

AAV-CMV-YFP-FKBP-18TM, AAV-CMV-YFP-FKBP and AAV-CMV-TOM20-CFP-FRB were produced based on AAV-DJ CMV Expression System (Cell Biolabs, VPK-410-DJ). HEK293T cells were seeded in two six-well cell culture plates (FALCON, 353046) at a density of $1.0 \times 10^6$ cells/well and next day transfected with *pAAV-DJ*, *pHelper* and *pAAVs* encoding transgenes. About 30 min before transfection, medium change was carried out with 5% FBS/DMEM supplemented with 1% Pen-Strep. For one well, 3 μg of a mixture of 1 μg of *pAAV-DJ*, 1 μg of *pHelper*, and 1 μg of *pAAVs* encoding transgenes were used. Six micrograms of PEI-Max (Polysciences, #24765-1) were used for transfection. After 24 h post-transfection, the culture medium was changed with 1% FBS/DMEM supplemented with 1% Pen-Strep. AAV purification was performed according to previous reports[56,57]. Transfected cells and medium were harvested in one plastic tube 72 h after transfection. Three milliliters of chloroform were added to the cell suspension (Sigma, 288306-100 ML). To extract AAV particles, the solution was thoroughly vortexed for 5 min. Cell debris were removed by centrifugation (5 min, $3000 \times g$, 4 °C). AAV particles were precipitated by adding 9.4 mL of 50% (v/v) of Polyethylene glycol (PEG) and 7.6 ml of 5 M sodium chloride (VWR, SS0430-500GR) and incubating for 2 h on ice. AAV precipitates were collected by centrifugation (30 min, $3000 \times g$, 4 °C). The supernatant was completely discarded and pellets were dissolved with 1.4 ml of PBS. To remove contaminant DNA and RNA, 3.5 μl of 1 M MgCl$_2$ (Sigma, M1028), 1.4 μl of DNase I (ThermoFisher, EN0521) and 1.4 μl of 10 μg/μl RNase A (Sigma 10109142001) were added and incubated at 37 °C for 20 min. To remove DNase and RNase, chloroform was applied and solution was thoroughly vortexed for 1 min. By the subsequent centrifugation (5 min, $12,500 \times g$, room temperature), cell debris was removed and supernatant were collected. This chloroform treatment was repeated at least 3 times. The resultant solution was concentrated in PBS using Amicon ultra centrifugal filter (Millipore, UFC510024). For titration, AAV preparations were treated with 1 unit of DNase I (ThermoFisher, EN0521) at 37 °C for 15 min, then incubated at 95 °C for 2 min to inactivate DNase I and finally incubated in 1% SDS at 70 °C for 15 min to lyse AAV particles. After optimal dilution of this solution, viral genome copies were quantified by qPCR using SsoAdvanced Universal SYBR Green Supermix (Bio-Rad, 1725272) and primers for a CMV promoter (Forward: 5′-ATAACT-TACGGTAAATGGCCCGCC-3′, reverse: 5′-ACTCCACCCATTGACGT-CAATGGA-3′)

## Culture and infection of primary mouse hippocampal neurons

Hippocampal neurons from embryonic day 18 (E18) C57/BL6 mice (Charles River) were prepared from embryos of unidentified sex were used as previously described[58]. In brief, hippocampi were dissected and incubated in papain for 25 min, then triturated and plated on poly-ʟ-lysine-coated 8-well glass chambers (Sigma, P2636, Cellvis, C8-1-N) at a density of 25,000 cells/well in 5% horse serum (Gibco 26050-088) containing Neurobasal medium (Gibco 21103-049) supplemented with 2% B-27 (Gibco 17504-044), 2 mM GlutaMax (Gibco 35050-061), and 100 U/ml Penicillin-Streptomycin (Gibco 15140-122). At DIV1, the medium was switched to Neurobasal medium with 2% B-27 and 2 mM GlutaMax, and neurons were thereafter maintained in this medium. The cultures were stored in a 37 °C incubator with 5% CO$_2$ until the day of experiments. For the gene transduction into primary neurons, AAVs were infected at MOI = 40,000. Neurons were analyzed 2 days after infection.

## Live-cell imaging

Twenty thousand cells/well for HeLa cells, 40,000 cells/well for COS-7 cells and 30,000 cells/well for MEFs (Wild-type or *Drp*1 KO) were seeded into poly-ᴅ-lysine-coated 8-well glass chamber (Sigma, P6407-5MG, Cellvis, C8-1-N). The next day, cells were transfected with

mCherry-, YFP-, or HaloTag-FKBP-PLAAT (including PLAAT1,2, 3, 4, 4dTM, 5, and PLAAT3 mutants) plasmids and FRB-anchor plasmids. The transfection was performed with FuGENE HD or XtremeGENE9 (Sigma, 6365787001) according to the manufacturer's protocol. Then cells were incubated for 15–24 h before live-cell imaging. For live-cell fluorescence imaging, an Eclipse Ti inverted fluorescence microscope equipped with ×60 (for peroxisome observation, Nikon) or ×100 (the other observation, Nikon) oil-immersion objective lens and pco.edge sCMOS camera (for peroxisome observation, PCO) or Zyla 4.2 plus CMOS camera (for the other observation, Andor) was used. Time-lapse imaging was conducted at 37 °C, 5% $CO_2$ and humidity using a stage top incubator (Tokai Hit) and done at 0.5-1 min intervals for 60–90 min. For the induction of FKBP-FRB interaction, cells were treated at a final concentration of 100 nM rapamycin. Fluorescence intensities were measured with NIS element software (Nikon) or MetaMorph imaging software. For optical control of inducible 18TM recruitment, COS-7 cells were transfected with plasmids expressing HaloTag-mSspB-18TM (or HaloTag-mSspB-18TM-LD), iLID-MoA and TOM20-mCherry. Next day, to visualize Halo-Tag-conjugated proteins, JF646-conjugated Halo Tag ligand (Promega, GA1120) was added to culture at a final concentration of 200 nM and cells were incubated for 30 min before live-imaging. Images were taken using an Eclipse Ti with similar settings. After the first frame of images was taken, blue light (447 nm) started to be irradiated at approximately 1 min interval for 1 millisecond at the lowest stimulus intensity. Imaging was performed for about 90 min. For quantification, cells with or without fully deformed mitochondria were manually counted using images of cells irradiated for 90 min.

### Observation of peroxisome reduction following PLAAT3 expression

To observe changes of peroxisomal numbers, COS-7 cells were transfected with plasmids expressing YFP, YFP-FL or YFP-18TM and mSca-Peroxi. After 48 h, imaging was carried out using Eclipse Ti. Peroxisome puncta were manually counted and the fraction of cells with <20 peroxisomes was calculated.

### Quantification of mitochondrial deformation and PLAAT3 translocation

For mitochondrial deformation, COS-7 cells were transfected with plasmids encoding mCherry-FKBP-PLAAT3 or -PLAAT3-mutants and CFP-FRB-MoA. Next day, transfected cells were treated with rapamycin for 50 min. Cell images were taken by ×100 oil-immersion objective lens and Eclipse Ti, 93-138 cells from three different experiments were analyzed. Cells with or without fully deformed mitochondria were manually counted based on which their fractions were calculated.

### Stress- or membrane-damage-induced recruitment of PLAAT3 FL and mutants

To quantify translocation of FL and 18TM after treatment with agents inducing membrane damage, oxidative stress and hyperosmotic stress, we carried out transfection into COS-7 cells with plasmids encoding mCherry-FKBP, mCherry-FKBP-FL or mCherry-FKBP-18TM (or their YFP versions and LD versions) and CFP-FRB-MoA. Next day, 1 mM of LLOME (Cayman, 16008) was applied to induce membrane damage. For inducing oxidative stress, 100 μM and 500 μM of $H_2O_2$ (Sigma, H1009) were added to culture. For inducing hyperosmotic stress, 200 mM and 500 mM of Sucrose (Sigma, S0389) or 50 mM and 100 mM of NaCl (Sigma, S3014) were applied to culture. These reagents were diluted in 10% FBS/DMEM supplemented with 1% Pen-Strep. One (for LLOME treatment) or two (for treatment with the other agents) h after addition, cell images were taken using ×60 oil-immersion objective lens and Eclipse Ti and 305-405 cells from three different experiments were analyzed. The proportion of cells with

merged signals between YFP or mCherry and CFP-FRB-MoA was manually counted.

### Imaging analysis

For co-localization analysis, COS-7 cells were transfected with plasmids encoding YFP-, YFP-FL, YFP-18TM (or their mCherry versions) and organelle marker proteins: PEX3-YFP for Peroxisomes, CFP-MoA for mitochondria, CFP-SEC61B for ER, Giantin-CFP for Golgi, LAMP1-CFP for lysosomes, mCherry-LC3 for autophagosomes, mCherry-Rab5 for endosomes, and mCherry-Lamin A for nucleus. In particular, to clearly see autophagosomes, cells were treated with 100 μM of chloroquine for one overnight. Next day, cells were imaged. To quantitatively analyze co-localization, line scanning analysis was performed using NIS element (Nikon), and correlation coefficients were calculated with MetaMorph imaging software. For the intensity analysis of fluorescent proteins and TMRE, regions of interest (ROIs) were manually drawn on analyzed organelles guided by anchor protein fluorescence. Using ROIs, the fluorescence intensities of each color channel were measured using NIS element. For membrane potential measurement using TMRE, COS-7 cells were transfected with plasmids encoding YFP-FKBP-18TM (or YFP-FKBP-18TM-LD) and CFP-FRB-MoA, and next day, were treated with 35 nM TMRE (final concentration) for 20 min before live imaging. Imaging was carried out for 90 min. Time-lapse fluorescence intensities were quantified with NIS element software (Nikon). In quantification of these experiments, the intensities were normalized by the intensity from the first frame. As a control, CCCP [Carbonyl cyanide 4-(trifluoromethoxy) phenylhydrazone] was added to culture at a final concentration of 10 μM and we confirmed that TMRE signals were quickly lost within 5 min after the treatment. To examine the dependency of loss of membrane potential on mPTP opening, cells were treated with 1 mM CsA for 1 h before imaging. After rapamycin treatment for 105 min, cells were imaged by Eclipse Ti. While imaging, CsA was not removed. COS-7 cells were also transfected with plasmids encoding YFP-FKBP-18TM, mCherry-MoA, and Su9-CFP. Next day, live-cell imaging was performed. To examine specificity of an 18TM operation to target organelles, we used plasmids expressing YFP-FKBP18TM, CFP-FRB-MoA or PEX3-CFP-FRB and mSca-Peroxi and induced deformation of mitochondria or peroxisomes by treating cells with rapamycin for 30 min. In mitochondria deformation cells, we manually counted mSca-Peroxi⁺ puncta at 0 and 30 min after rapamycin treatment. In PEX3-CFP-FRB-expressing cells, we performed immunostaining with an anti-catalase antibody. Correlation coefficient values between PEX3-CFP-FRB signals and catalase signals using MetaMorph imaging software. In cells expressing YFP-FKBP-18TM and CFP-FRB-MoA, we performed immunostaining with an anti-TOM20 antibody 0 and 60 min after rapamycin treatment. Correlation coefficient values were also calculated with MetaMorph imaging software. To quantitatively evaluate mSca-Peroxi leakage, HeLa cells were transfected with YFP-FKBP-18TM (18TM-LD), PEX3-CFP-FRB and mSca-Peroxi. For quantification of cytosolic mSca-Peroxi, transfected cells were treated for 30 min. ROIs were selected in cytosol without peroxisome puncta and their signal intensities were measured with MetaMorph imaging software. Additionally, using the same culture and transfection condition, we quantified the number of mSca-peroxi⁺ puncta of cells treated with rapamycin for 0 and 30 min. mSca-Peroxi⁺ puncta were manually counted. To quantify the percentage of catalase-negative cells, cells expressing both YFP-FKBP-18TM (or YFP-18TM-LD) and PEX3-CFP-FRB were selected. Because overexpression of PEX3-CFP-FRB at the highest level induced its mislocalization to mitochondria, the cells showing mislocalized PEX3-CFP-FRB were excluded from the analysis. The analyzed cells without colocalization between PEX3 and endogenous catalase were counted as catalase-negative cells. The intensities were normalized as the intensity at $t = 0$ to 1. Cells (111-121 cells) were chosen from three different experiments. In these

experiments, to reproducibly observe protein leakage, we selected and imaged cells with higher expression levels of both 18TM and anchor proteins.

### ATP FRET measurement

FRET signal was measured by exciting the mitAT1.03 with CFP excitation light and collecting emission with YFP emission filter at 37 °C. FRET images were thresholded to remove background before any contrast adjustments. To measure a FRET value over CFP intensity (referred to as FRET/CFP), we chose three independent regions in each organelle based on the YFP images, and then FRET and CFP intensity values in the corresponding region was automatically calculated by MetaMorph 7.8 imaging software. The average of FRET/CFP was calculated from at least three independent experiments.

### Electron microscopy

For correlative EM, HeLa cells were cultured on sapphire disks carbon-coated with a grid pattern. This pattern was used to locate the region of interest in an electron microscope. Following poly-L-lysine coating, cells were cultured overnight on the sapphire disks in an 8-well glass chamber before the transfection. Around 24 h after transfection, the PLAAT recruitment was induced for 30 min with 100 nM rapamycin, and cells fixed with 4% paraformaldehyde in PBS for 15 min. Cells were subsequently washed with PBS three times and then imaged using an Eclipse Ti2 inverted fluorescence microscope (Nikon) equipped with ×10 and ×60 objective lens and ORCA-Fusion BT Digital CMOS camera (Hamamatsu). Bright field images of the carbon grid pattern on the sapphire disks were simultaneously obtained to locate cells-of-interest in later steps. Following fluorescence imaging, cells were further processed for EM. Cells were fixed with 2% glutaraldehyde, 1 mM $CaCl_2$, and 0.1 M cacodylate buffer, pH 7.4, for 1 h on ice. After washes in 0.1 M cacodylate buffer, samples were post-fixed in 1% $OsO_4$, 1% potassium ferrocyanide and 0.1 M cacodylate buffer, for 1 h on ice. After washes in water, samples were incubated in 2% uranyl acetate for 30 min. After dehydration with 50, 70, 90, and 100% ethanol, samples were embedded into epon-araldite resin and cured for 72 h at 60 °C. The regions of interest were located based on the grid pattern. The plastic block was trimmed to the regions of interest and sectioned on a diamond knife using an ultramicrotome (Leica UC7). Approximately 40 consecutive sections (40 nm each) were collected onto the pioloform-coated grids and post-stained with 2.5% uranyl acetate in 50% methanol. Samples were observed using a H-7600 transmission electron microscope (Hitachi) equipped with a dual CCD camera (Advanced Microscopy Techniques). The fluorescence and electron micrographs were roughly aligned based on the carbon-coated grid patterns. The alignment was slightly adjusted based on the visible morphological features. Images were processed with FIJI software and Photoshop 2022 (Adobe). Partially overlapping image tiles were combined to produce a single image of a wide sectional area. The brightness and contrast were adjusted with FIJI software and Photoshop 2022 (Adobe) for the presentation. The raw images are available upon request.

### Immunofluorescence

HeLa and COS-7 cells were seeded at a density of 20,000 cells/well into poly-D-lysine-coated 8-well glass chamber. The next day, cells were transfected with appropriate plasmids such as those encoding YFP-FKBP-PLAAT and PEX3-mCeru3-FRB, or CFP-FRB-MoA and YFP-FKBP-18TM. The transfection was performed with FuGENE HD according to the manufacturer's protocol. After 24 h post-transfection cells were treated with 100 nM rapamycin or DMSO for 2 h. Then cells were washed with PBS (not containing $Mg^{2+}$ and $Ca^{2+}$) thrice and fixed with 4% paraformaldehyde (PFA, Electron Microscopy Sciences, 15714) in PBS for 10 min at room temperature. Next, cells were washed with PBS thrice and permeabilized and blocked with 2% bovine serum albumin and 0.1% Triton X-100 in PBS for 1 h at room temperature. After

blocking, cells were incubated with primary antibody against Catalase (Cell Signaling Technology, D4P7B, 1:1000) or TOM20 (Proteintech, 11802-1-AP) diluted in blocking buffer overnight at 4 °C. Cells were then washed with PBS thrice and incubated with diluted anti-rabbit secondary antibody conjugated with Alexa Fluor 594 (Thermo Fisher Scientific, A-11072) or Alexa Fluor 647 (Thermo Fisher Scientific, a-21244) at 1:1000 for 1 h at room temperature. Cells were washed with PBS thrice again and subjected to fluorescence imaging. The fluorescence imaging was performed on an Eclipse Ti inverted fluorescence microscope equipped with ×60 oil-immersion objective lens and pco.edge sCMOS camera.

### Colocalization analysis

Colocalization analyses were performed using a MetaMorph plugin function, CORRPLOT, which measures the correlation between the intensities of corresponding pixels in two fluorescence images and provides its correlation coefficient.

### Statistic and reproducibility

Statistical parameters including the definition and exact values of $n$ (number of cells and experiments) and deviation are described in figures and corresponding legends. Data are represented as mean ± s.d. $p$ values less than 0.05 were considered as statistically significant. Paired and unpaired two-tailed $t$-test as well as one-way/two-way ANOVA with Dunnett's multiple comparison tests were performed. Graphs were generated using GraphPad Prism 9 or Microsoft Excel.

### Reporting summary

Further information on research design is available in the Nature Research Reporting Summary linked to this article.

## Data availability

Source data are provided with this paper. All data supporting the conclusion in this manuscript are provided in the main article file, supplementary files or source data file.

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

## Acknowledgements

The authors would like to thank Hiromi Sesaki for *Drp1* knockout cells and constructs encoding Parkin and Su9, Hideaki Matsubayashi for fruitful discussion, and Hideki Nakamura for technical advice and construct sharing. We also thank the following researchers for constructive discussions on various aspects of the project: Noboru Mizushima, Hideaki Morishita, Tomoya Eguchi, Yukio Fujiki, Kanji Okumoto, Yuuta Imoto, Steve Gould, Michael Wolfgang, Marie Hardwick, Dwight Bergles. We also thank Robert DeRose and Willow Rock for proofreading the manuscript. We acknowledge technical assistance from Divisions of Research Instrument and Equipment and Radioisotope Research, Life Science Research Center, Kagawa University. We extend our appreciation to Junichi Takagi for kind support of manuscript preparation, and Takeharu Nagai for image analysis. This work was supported by discretionary funds to T.I., the National Institutes of Health (R01GM136858 to T.I., 1DP2 NS111133-01 to Sh.W.), the Chang-Zuckerberg Initiative to T.I. and Sh.W., Strategic Research Support Fund of Kagawa University Research Promotion Program 2021 (KURPP) to T.U., Charitable Trust MIU Foundation Memorial Fund to T.U., World Premier International Research Center Initiative (WPI), MEXT, Japan to S.T., and Takeda Science Foundation to T.U. Sh.W. is an Alfred P. Sloan fellow, a McKnight Foundation Scholar and a Klingenstein and Simons Foundation scholar. Sa.W. was supported by a postdoctoral fellowship from the Uehara Memorial Foundation.

## Author contributions

Sa.W. conceived and designed the research and performed all experiments unless otherwise mentioned. Sa.W. and T.I. wrote the original and revised manuscript. T.I. arranged collaborations, and performed some of the revision experiments. Y.N. performed the peroxisome experiments and contributed to manuscript preparation. Sh.W. and K.I. prepared the primary neuron cultures and performed EM and CLEM analysis. T.U. performed enzyme activity assay. S.T. performed flow cytometric viability assays.

## Competing interests

The authors declare no competing interests.
