## [Peer Review File · Nature Communications]

Reviewers' Comments:

Reviewer #1:

Remarks to the Author:

In their manuscript, Watanabe et al. develop a generalizable, controllable system to incapacitate membrane surrounded organelles, with the aim to allow studying individual organelle dysfunction and its effects in cells without major side effects on other organelles or cellular functions. To develop this system, the authors make use of the PLAAT family members of phospholipases, in particular PLAAT3, which has recently been described to induce broad, autophagy-independent organelle degradation during development of the lens in zebrafish and mice. To allow temporal and spatial control of the system, the authors employ an FKBP-FRB-based CID system, in which dimerization is controlled by addition of rapamycin. Watanabe et al. demonstrate that recruitment of PLAAT3 to a mitochondrial anchor rapidly induces swelling of mitochondria in a phospholipase activity-dependent manner. The authors continue to optimize the PLAAT3-based tool by eliminating unwanted characteristics of the full-length protein by generation of a series of mutants, resulting in the generation of a protein variant referred to as 18TM. Using this tool, the authors show that 18TM recruitment to mitochondria results in a loss of mitochondrial membrane potential as well as leakage of matrix proteins into the cytosol without induction of Parkin-dependent mitophagy. To underpin the generalizability of their tool, the authors further demonstrate that the 18TM system is compatible with photoactivatable dimerization and for viral transduction in cultured, murine primary neurons. Lastly, Watanabe et al. show that peroxisomal tethering of 18M impairs peroxisomal integrity.

General comments:

The current repertoire of methods to manipulate organelle function is limited to relatively slow genetic approaches and to pharmacological manipulations that are largely specific to distinct organelles, and thus, not readily applicable to other organelles. As such, there is indeed a need for a versatile, rapid and controllable tool that can be readily applied to different organelles and model systems without adverse effects on other organelles and overall cellular function/fitness.

The authors develop an elegant system that has this potential: It is easy to use, rapidly controllable and likely applicable to a number of organelles. However, their demonstration of the generalizable function and applicability of the system is limited to two organelles (namely mitochondria and peroxisomes), both of which have been shown to be targets/substrates of PLAAT3. In order to demonstrate the generalizability of this system, which is what should separate it from already existing methods, targeting to additional membrane-surrounded organelles would have been desirable.

Additionally, rapamycin-induced autophagy may have effects on organelle function and the implications of organelle damage. This may confound the observed results. The authors also demonstrate the applicability of alternative dimerization systems by using the iLID system, which is controlled by 450nm light and therefore circumvents the effects of rapamycin has on endogenous proteins. These data improve the broad applicability of the system and further allow reversible control of dimerization, although this was not demonstrated.

Specific points:

1. Line 128-132: Concerning their observation that PLAAT5 FL fails to induce mitochondrial deformation, the authors state that its TM "is shorter than the other PLAAT TMs by 2-3 amino acids" and that this suggests that besides its catalytic activity, the TM is crucial for this action. While the authors demonstrate that the TMs of PLAAT4 and PLAAT3 are crucial for the membrane deformation activity, the authors also show that shortening of the PLAAT3 TM by up to 3 amino acids does not abolish its function (as this is the 18TM construct that the rest of the manuscript focuses on). Thus, the argument of the shortened PLAAT5 TM domain is not sufficient to exclude that other factors or may be responsible for the difference between PLAAT5 and the other PLAAT family members.

2. Line 150-156: The authors mention in the introduction that when "a murine PLAAT3 truncation mutant that localizes to mitochondria was constitutively expressed in cultured cells, a series of mitochondrial defects were observed after two days, including swelling, fragmentation, membrane degradation, and loss of internal structures such as cristae" (Morishita et al. 2021). Expression of

the CT domain truncation mutants constructed in the present study resulted in mislocalization of mutant PLAAT3 to mitochondria or ER and induced mitochondrial swelling even in the absence of rapamycin treatment. It would be good to mention here that this is in line with previous data and cite the reference again.

3. Line 176-177: The direct connection between localization to peroxisomes and cell death is not clear. Has cell death been linked directly to the defective peroxisome biogenesis? In that case, a reference should be added here.

4. Figure 1e, f: PLAAT3 has previously been shown to negatively regulate viability of cancer cells, which the authors acknowledge. This is an important consideration given that many experiments are done in cultured human cancer cell lines. To assess cell viability upon expression of YFP fusions of wildtype FL, LD or 18TM PLAA3 constructs, the authors use COS-7 cells and analyze the percentage of YFP-fusion protein expressing cells 48h post transfection. This experiment raises two concerns: Firstly, COS-7 cells are a monkey-derived, fibroblast-like cell line. As the negative effect on viability has primarily been shown on cancer cells, it would have been favorable to perform this experiment in human cancer cell lines, which have been used in other experiments throughout the manuscript. Second, while this experiment may give some information on cell viability after transfection of the constructs, it is doubtful whether YFP-fluorescence should be directly translated into a measure of cell viability, especially using a single, end-timepoint measurement. Transfection efficiency as well as expression levels of the construct are greatly influencing this measurement without being directly linked to viability. Using this approach, the control, which expressed only YFP protein, displays a relative cell viability of 40% which will be confusing to the reader. Another assay should be used to monitor cell viability of transfected cells, specifically.

5. The authors demonstrate that 18TM targeting to mitochondria induces mitochondrial swelling, loss of membrane potential (TMRE) and leakage of matrix proteins (Su9), while it does not result in mitophagy as measured by recruitment, or lack thereof, of the ubiquitin ligase Parkin, which has previously been linked to loss of mitochondrial membrane potential. Additional experiments analyzing mitochondrial function, such as e.g. ATP synthesis, could therefore improve the understanding of the effects of the targeted phospholipase activity.

6. Figure 5: A merge of the channels would visualize the recruitment of both control and 18TM better than just split channels. While HaloTag-mSspB certainly works as a control, using a lipase dead version of 18TM, similar to all other experiments, would improve these data. Especially given the importance of these data for the generalizability and broad applicability of the system.

7. Figure 5a: In the timeline "minite" should be "minute".

8. Figure 5b: "HaloTag-mspB" should be "HaloTag-mSspB"

9. Line 295-297 and Figure 6a-d: The authors state that targeting 18TM to peroxisomes using a Pex3-based anchor leads to decrease of the fluorescent signal of the matrix protein mSca-Peroxi, while YFP-FKBP-18TM and PEX3-CFP-FRB remained constant. However, both the micrographs in 6a and the enlarged insets in 6c do not seem to suggest a decrease in PEX3 signal (and number of structures) over time in cells expressing phospholipase active 18TM protein. This should be addressed.

10. Figure 6e: Similar to other figures, it should be indicated where the inset is derived from in the image.

11. Line 319-321: The described findings have previously been described and references should be cited here in addition to the findings provided by the authors.

12. Line 366-368, the authors state that the expression levels of each of the components of the dimerization system are critically important for defunctionalization. This is very speculative as the authors have not provided data that support this statement.

13. Line 641: Cell line should be specified

14. Overall, the analysis and statistics section could be improved:

a. Specification of the method of data analysis (manual, semi-automated, automated) in the figure legends and methods section should be added

b. Quantification of time-lapse imaging data (Fig. 3a, 4d, e, 6d) should be improved by combining values obtained from several cells from different replicates (mean +/- s.d.) to demonstrate the reproducibility of this phenotype and data across populations and experiments rather than showing single cell data. Was the bleaching effect taken into account in 4d and 4e?

c. Fig 2c: The authors state that data are derived from 93 cells from three independent experiments. Standard deviation or error should be added to the bar graphs.

d. Fig. 3e: How was recruitment of Parkin determined? Significance/p-value could be added here.

e. Fig S3, S5: PCC and statistical analysis for colocalization are required.

f. Fig S6, quantification and statistical analysis are required.

g. Fig. S7, statistical analysis are required.

h. Fig S8, quantification and statistical analysis are required.

Reviewer #2:

Remarks to the Author:

PLAAT3 is a member of the phospholipase/acyltransferase (PLAAT) family and functions as a phospholipase A1/A2 enzyme. Earlier, Uyama et al. showed that the overexpression of PLAAT3 in mammalian cells dissipates peroxisomes (ref. 24 and 25 in this manuscript). In 2021, Morishita et al. reported that PLAAT3 degrades organelle membranes during development of lens transparency (17).

In the present study, the authors report that a truncation mutant of PLAAT3 called "18TM", which is still catalytically active but shows intracellular localization different from that of full-length PLAAT3, is a useful tool for rapid defunctionalization of intracellular organelles such as mitochondria and peroxisomes. This novel strategy may contribute to further studies on intracellular organelles. The methodology is sound. I just have some comments.

Major comments:

1. Although the authors use catalytically inactive C113S mutants as negative control, actual PLA1/A2 enzyme activity of a series of the truncated mutant proteins should be measured. In particular, it is indispensable to show how much 18TM mutant retains enzyme activity comparing with full-length PLAAT3.

Minor comments:

1. Since Ref. 24 and 25 are important as a cue of the present strategy, these articles should be cited in the Introduction and the first paragraph of Discussion. The authors may also cite a recent review article on PLAAT family proteins such as "An involvement of phospholipase A/acyltransferase family proteins in peroxisome regulation and plasmalogen metabolism" (FEBS Lett 591, 2745-60, 2017).

2. In Fig. 1f, the cell viability of YFP as a negative control is only 43.6%. Is YFP itself toxic? Or is the transfection method used toxic?

Reviewer #3:

Remarks to the Author:

Major revision

In this manuscript, Watanabe et al. report that they developed a molecular tool to induce the defunctionalization of specific organelles via remodeling of their membrane phospholipids. The authors generated truncated phospholipase A/acyltransferases (PLAATs) targeted to specific organelle membranes using chemical- or light-triggered hetero dimerization systems (FKBP/FRB, and iLID/SspB). PLAAT induced mitochondrial morphological changes and depolarization. The concept of rapid defunctionalization of organelles via remodeling of the membrane phospholipids with engineered PLAAT is novel and could potentially be useful to elucidate the physiological roles of organelles in dynamic cellular processes. However, the manuscript did not adequately describe the target-specificity of PLAAT tools. Mislocalized PLAAT tools may induce unintentional defunctionalization of nontargeted organelles.

The experimental data also do not fully support the conclusions because of inappropriate statistical analyses and organelle labeling. Therefore, this manuscript requires major revisions to validate and clarify the following specific points.

Specific points

1. The target-specificity of PLAAT tools is unclear in the manuscript. The authors shall show that the glycerophospholipids in the targeted organelle membrane are exclusively altered by PLAAT. In addition, the manuscript requires electron microscopy of the whole cells to show the integrity of nontargeted organelles. These experimental results would strengthen the conclusions that PLAAT remodel the target membrane phospholipids and induce defunctionalization of organelles.
2. In Figures 1 and 2, Authors aim to show that mitochondrially targeted PLAAT tool induce mitochondrial morphological alteration. However, the detection method of mitochondrial morphology is not appropriate. Authors use as probe CFP-FRB-MoA, one component of the PLAAT tool. Authors shall detect mitochondrial morphology by immunostaining of endogenous mitochondrial markers and electron microscopy to eliminate the possibility that only the localization pattern of CFP-FRB-MoA changed, but not the mitochondrial morphology.
3. In Figures 1e and f, Authors aim to show viability of the cells transfected with PLAAT3 constructs by counting the number of Hoechst positive cells. However, this cell viability assay is inappropriate because it does not eliminate the possibility that differential expression levels of the constructs affect the results. The authors shall perform AnnexinV-PI apoptosis assay in YFP-positive cells to clarify that LD and 18TM do not induce apoptosis.
4. In Figures 1f and 6f, Authors inappropriately compare more than 3 groups by repetitive Student's t test. The authors shall perform multiple comparison tests to avoid type 1 errors.
5. In Figure 4 and Figure 6, Authors show decreased signals of matrix proteins in mitochondria and peroxisomes following the recruitments of PLAAT tools and conclude that these matrix proteins leaked into the cytosol. However, cytosolic signals of leaked proteins seems not to be increased. The authors shall perform subcellular fractionation to clarify whether matrix proteins leaked into the cytosol.
6. In Figures 5a and b, Authors aim to show the optogenetic operation of organelle defunctionalization with 18TM fused to a Halo Tag and mSspB (HaloTag-mSspB-18TM) and mitochondria-anchored iLID (iLID-MoA). However, the mitochondrial morphological alterations are not appreciable from the images. The authors shall show clearer images, movies, and quantification. Furthermore, Authors shall reproduce the mitochondrial defunctionalization by optogenetic PLAAT activation. Besides the defunctionalization of mitochondria in the whole cell, the experiments to show the perturbation of the focal site of mitochondria by the irradiation of blue light to a small area may support the conclusion of the system's usefulness.
7. In Figures 6a and b, mScarlet-SRL seems to be mistargeted to nuclei. The authors shall use other fluorescent peroxisomal matrix proteins in the experiments. Furthermore, only a single dot is

quantified to show the permeability of peroxisomal membranes in Figure 6d. Authors shall repeat the experiments with control (YFP-FKBP-18TM-LD) and perform statistical analyses. PEX3-CFP-FRB seems to be partially targeted to non-peroxisomal structures in the images of Figure 6. Authors shall immunostain endogenous peroxisomal membrane proteins and perform colocalization analysis. Moreover, electron microscopy is required to clarify whether the morphology of peroxisomes is not altered by the peroxisomal PLAAT tool.

8. lines 221 and 222, authors state: "We next examined if mitophagy took place following the 18TM-mediated mitochondria deformation." However, apart from the recruitment of YFP-Parkin to mitochondria, they did not examine the mitophagy per se. Therefore, Authors shall measure mitochondrial mass, mito colocalization with LC3, and mitophagic flux (ex. ratiometric imaging of mito-Keima).

REVIEWER COMMENTS

Reviewer #1 (Remarks to the Author):

In their manuscript, Watanabe et al. develop a generalizable, controllable system to incapacitate membrane surrounded organelles, with the aim to allow studying individual organelle dysfunction and its effects in cells without major side effects on other organelles or cellular functions.

To develop this system, the authors make use of the PLAAT family members of phospholipases, in particular PLAAT3, which has recently been described to induce broad, autophagy-independent organelle degradation during development of the lens in zebrafish and mice. To allow temporal and spatial control of the system, the authors employ an FKBP-FRB-based CID system, in which dimerization is controlled by addition of rapamycin. Watanabe et al. demonstrate that recruitment of PLAAT3 to a mitochondrial anchor rapidly induces swelling of mitochondria in a phospholipase activity-dependent manner. The authors continue to optimize the PLAAT3-based tool by eliminating unwanted characteristics of the full-length protein by generation of a series of mutants, resulting in the generation of a protein variant referred to as 18TM. Using this tool, the authors show that 18TM recruitment to mitochondria results in a loss of mitochondrial membrane potential as well as leakage of matrix proteins into the cytosol without induction of Parkin-dependent mitophagy. To underpin the generalizability of their tool, the authors further demonstrate that the 18TM system is compatible with photoactivatable dimerization and for viral transduction in cultured, murine primary neurons. Lastly, Watanabe et al. show that peroxisomal tethering of 18M impairs peroxisomal integrity.

General comments:

The current repertoire of methods to manipulate organelle function is limited to relatively slow genetic approaches and to pharmacological manipulations that are largely specific to distinct organelles, and thus, not readily applicable to other organelles. As such, there is indeed a need for a versatile, rapid and controllable tool that can be readily applied to different organelles and model systems without adverse effects on other organelles and overall cellular function/fitness.

The authors develop an elegant system that has this potential: It is easy to use, rapidly controllable and likely applicable to a number of organelles. However, their demonstration of the generalizable function and applicability of the system is limited to two organelles (namely mitochondria and peroxisomes), both of which have been shown to be targets/substrates of PLAAT3. In order to demonstrate the generalizability of this system, which is what should separate it from already existing methods, targeting to additional membrane-surrounded organelles would have been desirable.

Additionally, rapamycin-induced autophagy may have effects on organelle function and the implications of organelle damage. This may confound the observed results. The authors also demonstrate the applicability of alternative dimerization systems by using the iLID system, which is controlled by 450nm light and therefore circumvents the effects of rapamycin has on endogenous proteins. These data improve the broad applicability of

the system and further allow reversible control of dimerization, although this was not demonstrated.

We appreciate the reviewer for their deep understanding of our study accompanied by insightful comments. We cannot help but agree with the conclusion that the applicability of our PLAAT strategy should be reinforced by extending its use to other organelles as well as better establishing optogenetic control. Besides thoroughly addressing the specific points below, we revisited our optogenetic experiment in order to reassess with finer controls. In consideration of the opportunity to validate our system through perturbation of additional organelles, we have identified the resources and expertise available to the lab necessary to achieve this as a part of future exploration. To properly reflect our present dataset demonstrating defunctionalization of two specific organelles, we made a change to the manuscript title as follows: *“Defunctionalizing intracellular organelles such as mitochondria and peroxisomes with engineered phospholipase A/acyltransferases (PLAATs)”*.

Specific points:

1. Line 128-132: Concerning their observation that PLAAT5 FL fails to induce mitochondrial deformation, the authors state that its TM “is shorter than the other PLAAT TMs by 2-3 amino acids” and that this suggests that besides its catalytic activity, the TM is crucial for this action. While the authors demonstrate that the TMs of PLAAT4 and PLAAT3 are crucial for the membrane deformation activity, the authors also show that shortening of the PLAAT3 TM by up to 3 amino acids does not abolish its function (as this is the 18TM construct that the rest of the manuscript focuses on). Thus, the argument of the shortened PLAAT5 TM domain is not sufficient to exclude that other factors or may be responsible for the difference between PLAAT5 and the other PLAAT family members.

We admit that our original argument of the inability of PLAAT5 to induce organelle deformation was not reasonable. To amend this, we newly explored the predicted secondary structure of PLAAT5 further by using two additional programs (SOSUI and TMHMM) which consistently indicate that PLAAT5 is devoid of a predicted TM domain at the C-terminus, unlike other PLAAT members. Therefore, we now speculate that the reason why PLAAT5 did not show mitochondrial deformation is due to the missing TM domain. The corresponding main text is accordingly revised as follows:

“A major difference between PLAAT5 and other members is the absence of a predicted TM domain; SOSUI and TMHMM do not predict PLAAT5 to contain a TM domain at the C-terminus (**Supplementary Fig. 1a**).” (Lines 124-126)

2. Line 150-156: The authors mention in the introduction that when “a murine PLAAT3 truncation mutant that localizes to mitochondria was constitutively expressed in cultured cells, a series of mitochondrial defects were observed after two days, including swelling, fragmentation, membrane degradation, and loss of internal structures such as cristae” (Morishita et al. 2021). Expression of the CT domain truncation mutants constructed in the present study resulted in mislocalization of mutant PLAAT3 to mitochondria or ER and

induced mitochondrial swelling even in the absence of rapamycin treatment. It would be good to mention here that this is in line with previous data and cite the reference again.

We followed the suggestion by adding the citation (Morishita et al., 2021) as follows:

“The observed mitochondrial mislocalization of human 2CT was consistent with a murine version of 2CT described previously²⁰.” (Lines: 151-152)

3. Line 176-177: The direct connection between localization to peroxisomes and cell death is not clear. Has cell death been linked directly to the defective peroxisome biogenesis? In that case, a reference should be added here.

We looked for such a reference to no avail, making us realize that we previously had an unfair assumption on their direct causal relationship. Therefore, we removed the misleading sentence.

4. Figure 1e, f: PLAAT3 has previously been shown to negatively regulate viability of cancer cells, which the authors acknowledge. This is an important consideration given that many experiments are done in cultured human cancer cell lines. To assess cell viability upon expression of YFP fusions of wildtype FL, LD or 18TM PLAA3 constructs, the authors use COS-7 cells and analyze the percentage of YFP-fusion protein expressing cells 48h post transfection. This experiment raises two concerns: Firstly, COS-7 cells are a monkey-derived, fibroblast-like cell line. As the negative effect on viability has primarily been shown on cancer cells, it would have been favorable to perform this experiment in human cancer cell lines, which have been used in other experiments throughout the manuscript. Second, while this experiment may give some information on cell viability after transfection of the constructs, it is doubtful whether YFP-fluorescence should be directly translated into a measure of cell viability, especially using a single, end-timepoint measurement. Transfection efficiency as well as expression levels of the construct are greatly influencing this measurement without being directly linked to viability. Using this approach, the control, which expressed only YFP protein, displays a relative cell viability of 40% which will be confusing to the reader. Another assay should be used to monitor cell viability of transfected cells, specifically.

Following these helpful suggestions, we used HeLa cancer cells and analyzed a fraction of YFP expressing cells at three different time points of post-transfection using a flow cytometer. As a result, all constructs we tested, namely YFP, YFP-FL, YFP-FL-LD- and YFP-18TM, indicated comparable protein expression on days 1 and 3, while cells expressing YFP-FL decreased significantly on day 5 (~70% reduction), confirming the toxicity of the full length PLAAT3. Importantly, this experiment confirmed minimal cytotoxicity of PLAAT3-18TM (now shown in **Fig. 1g**).

5. The authors demonstrate that 18TM targeting to mitochondria induces mitochondrial swelling, loss of membrane potential (TMRE) and leakage of matrix proteins (Su9), while it does not result in mitophagy as measured by recruitment, or lack thereof, of the ubiquitin ligase Parkin, which has previously been linked to loss of mitochondrial membrane

potential. Additional experiments analyzing mitochondrial function, such as e.g. ATP synthesis, could therefore improve the understanding of the effects of the targeted phospholipase activity.

We performed a new experiment and examined ATP synthesis using a mitAT1.03 FRET biosensor in cells undergoing 18TM-mediated mitochondrial defunctionalization, along with proper controls. As a result, we observed 17% decrease after rapamycin addition for two hours, compared to cells treated with DMSO (hence no 18TM recruitment) (Fig. 3d, e).

6. Figure 5: A merge of the channels would visualize the recruitment of both control and 18TM better than just split channels. While HaloTag-mSspB certainly works as a control, using a lipase dead version of 18TM, similar to all other experiments, would improve these data. Especially given the importance of these data for the generalizability and broad applicability of the system.

According to the reviewer's suggestion, we added merged images, and performed proper control experiments with lipase-dead version of 18TM (Fig. 5a, b). To further improve visibility of optogenetic recruitment of 18TM and ensuing morphological changes, we additionally included enlarged images in insets, besides quantifying the phenotypes (Fig. 5c).

7. Figure 5a: In the timeline “minite” should be “minute”.

8. Figure 5b: “HaloTag-mspB” should be “HaloTag-mSspB”

These typos are now corrected.

9. Line 295-297 and Figure 6a-d: The authors state that targeting 18TM to peroxisomes using a Pex3-based anchor leads to decrease of the fluorescent signal of the matrix protein mSca-Peroxi, while YFP-FKBP-18TM and PEX3-CFP-FRB remained constant. However, both the micrographs in 6a and the enlarged insets in 6c do not seem to suggest a decrease in PEX3 signal (and number of structures) over time in cells expressing phospholipase active 18TM protein. This should be addressed.

Our experiments shown in Fig. 6 indicated a peroxisomal signal decrease of luminal proteins such as mSca-peroxi and catalases, but not PEX3. Since PEX3, unlike mSca-peroxi and catalases, is a membrane protein, we concluded that membrane integrity was only partially lost such that these luminal proteins can leak out while membrane proteins can be retained. To back up these observations, we newly quantified their signal intensities in peroxisomes as well as a number of peroxisomes labeled with mSca-Peroxi or PEX-CFP-FRB (Fig. 6e,f).

10. Figure 6e: Similar to other figures, it should be indicated where the inset is derived from in the image.

As suggested, we now indicate where the inset is derived from (now Fig. 6g), and also other figures where applicable.

11. Line 319-321: The described findings have previously been described and references should be cited here in addition to the findings provided by the authors.

As pointed out, we revised sentences and added citations to clarify that these findings are consistent with previous reports. The corresponding main text now reads as follows:

“Consistent with previous reports^{17-19,27,28}, full-length PLAAT3 localizes to peroxisomes (Supplementary Figure 3), impairs their biogenesis (Supplementary Figure 6), and, when overexpressed, compromises cell viability (Fig. 1).” (Lines 389-391)

12. Line 366-368, the authors state that the expression levels of each of the components of the dimerization system are critically important for defunctionalization. This is very speculative as the authors have not provided data that support this statement.

Indeed, this statement is based on our empirical observation and not data-supported. We thus decided to remove the corresponding discussion.

13. Line 641: Cell line should be specified

We now specify cell lines in the figure legends.

14. Overall, the analysis and statistics section could be improved:

a. Specification of the method of data analysis (manual, semi-automated, automated) in the figure legends and methods section should be added

As suggested, we now mention analysis methods in Legends and Methods.

b. Quantification of time-lapse imaging data (Fig. 3a, 4d, e, 6d) should be improved by combining values obtained from several cells from different replicates (mean +/- s.d.) to demonstrate the reproducibility of this phenotype and data across populations and experiments rather than showing single cell data. Was the bleaching effect taken into account in 4d and 4e?

We now indicate values from at least three replicates to compute mean \pm SD, which were followed by statistical analysis.

The fluorescence intensity values presented in Fig. 4d, e did not undergo photobleach correction. Of note, our live-cell imaging conditions including light intensity, exposure time, pixel binning is optimized for each experiment to minimize photobleaching (and other adverse effects including phototoxicity) prior to performing data collection.

c. Fig 2c: The authors state that data are derived from 93 cells from three independent experiments. Standard deviation or error should be added to the bar graphs.

To clarify this statement, we analyzed 36, 25, and 32 cells from experiments 1, 2 and 3, respectively. We now clearly state this and indicate mean and SD values in **Fig. 2c**.

d. Fig. 3e: How was recruitment of Parkin determined? Significance/p-value could be added here.

We manually counted cells indicating Parkin accumulation. Subcellular distribution of Parkin was unambiguously distinct when accumulated (as seen in the representative images in this figure). We now describe how we analyzed the data in the legend.

e. Fig S3, S5: PCC and statistical analysis for colocalization are required.

According to the reviewer's suggestion, we measured correlation coefficient values using MetaMorph and applied statistical analysis.

f. Fig S6, quantification and statistical analysis are required.

g. Fig. S7, statistical analysis are required.

h. Fig S8, quantification and statistical analysis are required.

As suggested, we have performed quantification and the statistical tests and now describe them in the corresponding figures, legends and Methods.

Reviewer #2 (Remarks to the Author):

PLAAT3 is a member of the phospholipase/acyltransferase (PLAAT) family and functions as a phospholipase A1/A2 enzyme. Earlier, Uyama et al. showed that the overexpression of PLAAT3 in mammalian cells dissipates peroxisomes (ref. 24 and 25 in this manuscript). In 2021, Morishita et al. reported that PLAAT3 degrades organelle membranes during development of lens transparency (17).

In the present study, the authors report that a truncation mutant of PLAAT3 called “18TM”, which is still catalytically active but shows intracellular localization different from that of full-length PLAAT3, is a useful tool for rapid defunctionalization of intracellular organelles such as mitochondria and peroxisomes. This novel strategy may contribute to further studies on intracellular organelles. The methodology is sound. I just have some comments.

Major comments:

1. Although the authors use catalytically inactive C113S mutants as negative control, actual PLA1/A2 enzyme activity of a series of the truncated mutant proteins should be measured. In particular, it is indispensable to show how much 18TM mutant retains enzyme activity comparing with full-length PLAAT3.

To achieve this informative, yet technically demanding experiment, we established a new collaboration with an expert of the *in vitro* measurement of PLAAT catalytic activity. To our enthusiasm, we now demonstrate that C113S (LD) has indeed little to no activity, while 18TM retains ~10% activity of the full length (Fig. 1E).

Minor comments:

1. Since Ref. 24 and 25 are important as a cue of the present strategy, these articles should be cited in the Introduction and the first paragraph of Discussion. The authors may also cite a recent review article on PLAAT family proteins such as “An involvement of phospholipase A/acyltransferase family proteins in peroxisome regulation and plasmalogen metabolism” (FEBS Lett 591, 2745-60, 2017).

These citations are now added at proper sites in the main text.

2. In Fig. 1f, the cell viability of YFP as a negative control is only 43.6%. Is YFP itself toxic? Or is the transfection method used toxic?

Based on this comment, we realized that the original Y-axis title “Cell Viability” was misleading. What we really quantified was the amount of YFP-positive cells divided by the amount of all cells in a given field of view. Thus, we revised the Y-axis title as “YFP-positive cells (% of cells in a culture well)”.

Reviewer #3 (Remarks to the Author):

Major revision

In this manuscript, Watanabe et al. report that they developed a molecular tool to induce the defunctionalization of specific organelles via remodeling of their membrane phospholipids. The authors generated truncated phospholipase A/acyltransferases (PLAATs) targeted to specific organelle membranes using chemical- or light-triggered hetero dimerization systems (FKBP/FRB, and iLID/SspB). PLAAT induced mitochondrial morphological changes and depolarization. The concept of rapid defunctionalization of organelles via remodeling of the membrane phospholipids with engineered PLAAT is novel and could potentially be useful to elucidate the physiological roles of organelles in dynamic cellular processes. However, the manuscript did not adequately describe the target-specificity of PLAAT tools. Mislocalized PLAAT tools may induce unintentional defunctionalization of nontargeted organelles.

The experimental data also do not fully support the conclusions because of inappropriate statistical analyses and organelle labeling. Therefore, this manuscript requires major revisions to validate and clarify the following specific points.

We appreciate that the reviewer mentions the novelty of our approach and professionally shares constructive feedback to improve the study. To address the concern of insufficient evidence for our conclusion, we performed a series of new experiments and applied proper statistical analyses as detailed below.

Specific points

1. The target-specificity of PLAAT tools is unclear in the manuscript. The authors shall show that the glycerophospholipids in the targeted organelle membrane are exclusively altered by PLAAT. In addition, the manuscript requires electron microscopy of the whole cells to show the integrity of nontargeted organelles. These experimental results would strengthen the conclusions that PLAAT remodel the target membrane phospholipids and induce defunctionalization of organelles.

We agree with these comments and thus performed new experiments, including visualization of mitochondria phospholipids and ultrastructural assessment of non-targeted organelles. In short, we now have direct evidence supporting PLAAT-mediated phospholipid remodeling in the mitochondria while largely sparing unintended organelles. Below summarize these two experiments:

1. Visualization of mitochondria phospholipids

To directly visualize the mitochondrial phospholipid modification after 18TM recruitment, we employed a fluorescence-conjugated phosphatidylserine (PS) where an NBD fluorescent dye is covalently attached to the Sn2 position. When introduced into cells, this molecule has been shown to localize to membranes of Golgi and mitochondria. We expected 18TM recruitment to lead to hydrolysis of PS, liberating an NBD-labeled fatty acid chain from the mitochondria membrane. By measuring NBD fluorescence at mitochondria

before and after the 18TM recruitment, we indeed could observe significantly reduced NBD signal (**Supplementary Figure 10**). All other control conditions did not lead to such NBD reduction.

2. Morphological assessment of non-targeted organelles using correlative EM and live-cell fluorescence imaging

To assess organelle specificity of the 18TM-mediated defunctionalization, we performed two new experiments: live-cell fluorescence imaging and correlative EM analysis. In live-cell fluorescence imaging, we investigated whether induction of mitochondrial defunctionalization affects peroxisomal properties such as retention of luminal proteins, and conversely, whether induction of peroxisomal defunctionalization affects mitochondrial properties including their morphology (**Supplementary Figure 13**). In short, we confirmed the high organelle specificity where mitochondria defunctionalization had little to no impact on peroxisomes, and vice versa. In correlative EM analyses, we obtained ultrastructural organelle images in the cell undergoing 18TM-mediated mitochondrial defunctionalization (**Fig. 4f,g,h**). While we could confirm deformation of mitochondria, we found no remarkable morphological change or disruption of membrane integrity in all other organelles.

Taken together, these new results indicate high target specificity of our PLAAT strategy, at least for the duration of our experiment (~30 minutes). As organelles interact with each other and exchange materials, it is possible that defunctionalization of a given organelle impacts other organelles after some time.

2. In Figures 1 and 2, Authors aim to show that mitochondrially targetted PLAAT tool induce mitochondrial morphological alteration. However, the detection method of mitochondrial morphology is not appropriate. Authors use as probe CFP-FRB-MoA, one component of the PLAAT tool. Authors shall detect mitochondrial morphology by immunostaining of endogenous mitochondrial markers and electron microscopy to eliminate the possibility that only the localization pattern of CFP-FRB-MoA changed, but not the mitochondrial morphology.

To address this, cells expressing YFP-FKBP-18TM and CFP-FRB-MoA were subjected to immunostaining using an antibody against endogenous TOM20, one of the major mitochondria outer membrane proteins. After 18TM recruitment, mitochondria expectedly underwent deformation which was also confirmed in the TOM20 staining (**Supplementary Figure 9**). In fact, we observed nearly perfect colocalization between CFP-FRB-MoA and TOM20, regardless of the rapamycin addition (correlation coefficients: 0.96-0.97).

3. In Figures 1e and f, Authors aim to show viability of the cells transfected with PLAAT3 constructs by counting the number of Hoechst positive cells. However, this cell viability assay is inappropriate because it does not eliminate the possibility that differential expression levels of the constructs affect the results. The authors shall perform Annexin-V-PI apoptosis assay in YFP-positive cells to clarify that LD and 18TM do not induce apoptosis.

As suggested, we performed a flowcytometric analysis of cells transfected either with YFP, YFP-FL, YFP-FL-LD or YFP-18TM over 5 days post-transfection. On day 1, the transfected fraction of cells was

comparable between each sample, suggesting similar transfection efficiency. On day 3, this fraction remained, indicating a minimal effect on cell viability for a short-term period. On day 5, however, a fraction of YFP-FL-positive cells dropped by 70% relative to the YFP control, while 18TM did not exhibit such a drop (**Fig. 1g**). These support our original conclusion that cytotoxicity of the full length PLAAT3 is improved in 18TM.

To determine if the observed fractional decrease of FL-expressing cells is due to necrotic or apoptotic cell death, we then labeled cells with propidium iodide or SYTOX. Somewhat unexpectedly, we detected no more signal in YFP-FL-transfected cells than in control YFP-transfected cells. We concluded that our assay does not have sufficient sensitivity in detecting these cell death markers, owing to cell death in the background likely caused by transient transfection. Gene introduction in a less toxic manner such as viral infection or establishment of a stable cell line would be required to discern a cause for reduction in FL-transfected cells.

4. In Figures 1f and 6f, Authors inappropriately compare more than 3 groups by repetitive Student's t test. The authors shall perform multiple comparison tests to avoid type 1 errors.

Thanks for pointing out this error. We have adapted one-way ANOVA with Dunnett's multiple comparison tests.

5. In Figure 4 and Figure 6, Authors show decreased signals of matrix proteins in mitochondria and peroxisomes following the recruitments of PLAAT tools and conclude that these matrix proteins leaked into the cytosol. However, cytosolic signals of leaked proteins seems not to be increased. The authors shall perform subcellular fractionation to clarify whether matrix proteins leaked into the cytosol.

Upon quantification, we actually do see modest increase in mSca-peroxi in the cytosol (**Fig. 6e**), but not so much in Su9 (**Supplementary Figure 12**). This may reflect our observation that loss of mSca-peroxi at peroxisomes appears to be much more intense than that of Su9 at mitochondria.

6. In Figures 5a and b, Authors aim to show the optogenetic operation of organelle defunctionalization with 18TM fused to a Halo Tag and mSspB (HaloTag-mSspB-18TM) and mitochondria-anchored iLID (iLID-MoA). However, the mitochondrial morphological alterations are not appreciable from the images. The authors shall show clearer images, movies, and quantification. Furthermore, Authors shall reproduce the mitochondrial defunctionalization by optogenetic PLAAT activation. Besides the defunctionalization of mitochondria in the whole cell, the experiments to show the perturbation of the focal site of mitochondria by the irradiation of blue light to a small area may support the conclusion of the system's usefulness.

We replaced these original images with new images that clearly indicate mitochondrial deformation (**Fig. 5a,b**) to which we also added their merged images as well as enlarged images, along with their quantitative analysis (**Fig. 5c**).

Besides fast and inert induction of protein-protein interactions, the power of optogenetics indeed resides in reversible and subcellular operations. We agree that demonstrating these features with the PLAAT tool would be a great addition. Therefore, we conducted suggested optogenetic experiments. Though reversible and local recruitment of 18TM was readily possible, we could not observe reversible morphological changes or subcellularly localized deformation. We believe that this is largely due to the irreversible nature of mitochondria deformation, and generally low efficiency of optogenetic dimerization (K_d in μM) compared to chemically inducible dimerization (K_d in low nM). By using other optogenetic dimerization pairs such as Magnets and Cry2/CIBN, these intricate operations may become possible.

7. In Figures 6a and b, mScarlet-SRL seems to be mistargeted to nuclei. The authors shall use other fluorescent peroxisomal matrix proteins in the experiments.

When overexpressed, mScarlet-SRL (mSca-peroxi) proteins in excess start to spill over into diffuse areas of the cell such as the cytosol and nucleus. This is part of the reason why we extended this type of experiment to endogenous luminal proteins such as catalases (**Fig. 6g,h**).

Furthermore, only a single dot is quantified to show the permeability of peroxisomal membranes in Figure 6d. Authors shall repeat the experiments with control (YFP-FKBP-18TM-LD) and perform statistical analyses.

As suggested, we repeated the experiments with a proper control and performed statistical analyses (**Fig. 6e,f**), which validated observed leakage of the luminal proteins upon 18TM recruitment.

PEX3-CFP-FRB seems to be partially targeted to non-peroxisomal structures in the images of Figure 6. Authors shall immunostain endogenous peroxisomal membrane proteins and perform colocalization analysis.

To quantitatively evaluate how well PEX3-CFP-FRB localizes to peroxisomes, we measured a correlation coefficient (CC) between PEX3-CFP-FRB and endogenous peroxisomal catalases. The calculated CC turns out to be very high (0.85 ± 0.01), thus confirming PEX3 as a qualified peroxisomal marker. Our epifluorescence microscopy inadvertently collects some of the fluorescence signals outside of a given focal plane. It is possible that peroxisomal signals from outside the plane, typically shown blurry, may have appeared to be “non-peroxisomal”.

Moreover, electron microscopy is required to clarify whether the morphology of peroxisomes is not altered by the peroxisomal PLAAT tool.

As suggested, we newly performed an EM ultrastructural analysis of peroxisomes upon PLAAT recruitment. Here, we first identified cells undergoing a loss of the peroxisomal mSca-SRL signal upon rapamycin-induced 18TM recruitment (**Fig. 6i,j**). In these cells, we then obtained EM images of peroxisomes. Upon inspection, there were no apparent ultrastructural features that differed from those of untransfected neighboring cells (**Fig. 6k**).

8. lines 221 and 222, authors state: “We next examined if mitophagy took place following the 18TM-mediated mitochondria deformation.” However, apart from the recruitment of YFP-Parkin to mitochondria, they did not examine the mitophagy per se. Therefore, Authors shall measure mitochondrial mass, mito colocalization with LC3, and mitophagic flux (ex. ratiometric imaging of mito-Keima).

To further test if mitochondria deformation coincides with mitophagy, we newly analyzed colocalization between deformed mitochondria and LC3 or lysosomes. At which point, we did not observe colocalization of deformed mitochondria with LC3 or lysosomes (LC3: -0.19 ± 0.03 , lysosomes: -0.20 ± 0.16), suggesting that 18TM-mediated deformed mitochondria are unlikely to be subjective to mitophagy, at least within 6 hours of the deformation.

Reviewers' Comments:

Reviewer #1:

Remarks to the Author:

The authors largely addressed my questions. The manuscript could be considered for publication.

One point remains to be addressed:

Quantification of time-lapse imaging data (Fig. 4d, e, 6d) should be improved by combining values obtained from several cells from different replicates (mean \pm s.d.) to demonstrate the reproducibility of this phenotype and data across populations and experiments rather than showing single cell data.

Reviewer #2:

Remarks to the Author:

The authors appropriately revised the manuscript in response to my concerns.

In the proof, 14 of 14C should be changed to superscript.

Reviewer #3:

Remarks to the Author:

I commend the authors for an exhaustive and convincing review. I am convinced that the truncated PLAAT introduced here can be a very useful tool to genetically modulate organellar function in cells.

REVIEWERS' COMMENTS

Reviewer #1 (Remarks to the Author):

The authors largely addressed my questions. The manuscript could be considered for publication. One point remains to be addressed:

Quantification of time-lapse imaging data (Fig. 4d, e, 6d) should be improved by combining values obtained from several cells from different replicates (mean \pm s.d.) to demonstrate the reproducibility of this phenotype and data across populations and experiments rather than showing single cell data.

Fluorescence signals of luminal proteins in individual organelles are too variable to analyze average values or to perform kinetic analysis of many cells. To make this point, we prepared sample kinetic plots of several organelles randomly chosen from two cells (left: mitochondria, right: peroxisomes). While all organelles indicate a decrease over time (thus loss of luminal proteins is robust among cells), they are highly heterogenous in terms of the timing and kinetics of the leakage. The fluorescence signal is also generally fluctuating, likely due to the active movement of organelles. The observed heterogeneity should originate from a combination of multiple factors including the expression level of 18TM, an extent and kinetics of rapamycin-mediated 18TM recruitment to organelles, mechanical properties of organelle membranes, etc. Accordingly, we determined that the kinetic analysis of cell populations cannot be readily or fairly done, and therefore decided to remove the original kinetic plots from the manuscript. Instead, we now relocate the previous Fig. S12 (quantification of luminal proteins at two fixed time points) to the main figure as new Fig. 4d,e.

Reviewer #2 (Remarks to the Author):

The authors appropriately revised the manuscript in response to my concerns. In the proof, 14 of 14C should be changed to superscript.

This typo is now corrected.

Reviewer #3 (Remarks to the Author):

I commend the authors for a exhaustive and convincing review. I am convinced that the truncated PLAAT

introduced here can be a very useful tool to genetically modulate organellar function in cells.

Thank you for taking time to comment on the quality and quantity of our revision, as well as the power of our molecular tools!